# *TextTeacher*: What Can Language Teach About Images?

**Tobias Christian Nauen** *tobias_christian.nauen@dfki.de*
*RPTU University Kaiserslautern-Landau*
*German Research Center for Artificial Intelligence (DFKI)*

**Stanislav Frolov** *stanislav.frolov@dfki.de*
*German Research Center for Artificial Intelligence (DFKI)*

**Brian B. Moser** *brian.moser@dfki.de*
*German Research Center for Artificial Intelligence (DFKI)*

**Federico Raue** *federico.raue@dfki.de*
*German Research Center for Artificial Intelligence (DFKI)*

**Ahmed Anwar** *ahmed.anwar@dfki.de*
*German Research Center for Artificial Intelligence (DFKI)*
*RPTU University Kaiserslautern-Landau*

**Andreas Dengel** *andreas.dengel@dfki.de*
*German Research Center for Artificial Intelligence (DFKI)*
*RPTU University Kaiserslautern-Landau*

**Reviewed on OpenReview:** *https://openreview.net/forum?id=Xwb0aEUwKh*

## Abstract

The platonic representation hypothesis suggests that sufficiently large models converge to a shared representation geometry, even across modalities. Motivated by this, we ask: Can the semantic knowledge of a language model efficiently improve a vision model? As an answer, we introduce *TextTeacher*, a simple auxiliary objective that injects text embeddings as additional information into image classification training. *TextTeacher* uses readily available image captions, a pre-trained and frozen text encoder, and a lightweight projection to produce semantic anchors that efficiently guide representations during training while leaving the inference-time model unchanged. On ImageNet with standard ViT backbones, *TextTeacher* improves accuracy by up to +2.7 percentage points (p.p.) and yields consistent transfer gains (on average +1.0 p.p.) under the same recipe and compute. It outperforms vision knowledge distillation, yielding more accuracy at a constant compute budget or similar accuracy, but 33% faster. Our analysis indicates that *TextTeacher* acts as a feature-space preconditioner, shaping deeper layers in the first stages of training, and aiding generalization by supplying complementary semantic cues. *TextTeacher* adds negligible overhead, requires no costly multimodal training of the target model and preserves the simplicity and latency of pure vision models.

Project page with code and captions: **https://nauen-it.de/publications/text-teacher**

## 1 Introduction

Image classification is a canonical problem in computer vision, powering medical decision support (Sanderson & Matuszewski, 2022; Vezakis et al., 2024), autonomous driving (Wang et al., 2023) and object-centric perception (Girshick et al., 2014; He et al., 2017; Carion et al., 2020). Over the last few years, the Transformer (Vaswani et al., 2017) has emerged as a unifying architecture across modalities, with Large

Language Models (LLMs) advancing natural language processing (Radford et al., 2018; Devlin et al., 2019; Radford et al., 2019; Raffel et al., 2020; Touvron et al., 2023) and Vision Transformers (ViT) pushing visual recognition (Dosovitskiy et al., 2021; Touvron et al., 2021; Caron et al., 2021; Touvron et al., 2022; Oquab et al., 2024). In parallel, multimodal models such as VLMs and contrastively trained image–text encoders have demonstrated impressive zero-shot and transfer capabilities (Radford et al., 2021; Jia et al., 2021; Liu et al., 2023; Grattafiori et al., 2024), but rely on massive web-scale pretraining on tightly coupled image–text pairs.

We draw inspiration from the *platonic representation hypothesis* (PRH) (Huh et al., 2024), which proposes that sufficiently large models tend to converge to a shared representation geometry even when trained on different modalities. While the PRH is formally expected to hold only for large models and datasets, we treat it as a motivating hypothesis suggesting that representations learned by a large model can be used to guide a significantly smaller model, even if the student uses a different modality and would be too small to "discover" the platonic representation independently. This leads us to the central question: *Can we leverage the semantic knowledge learned by a language model to efficiently train a vision model?* In particular, without resorting to compute expensive contrastive multimodal training of our target model.

This question is relevant in settings where collecting large image–text corpora is infeasible or where budget constraints preclude web-scale multimodal training; be it due to hardware cost, power consumption, or $CO_2$ footprint (Xu et al., 2023). In particular, these costs can multiply during scientific discovery, where models may be re-trained multiple times (Wang & Zhu, 2024). It is also relevant for real-time, on-device, or safety-critical and regulated environments, where models must remain lightweight during inference or adhere to strict design constraints (Rabe et al., 2021; Myllyaho et al., 2021; Burton & Herd, 2023). If the answer is yes, we could tap into the rich conceptual structure already distilled in language models, leveraging the insight that large models can learn a shared geometry in a co-learning setting to guide visual learning *during training*, yet deploy an unimodal vision model *at test time*.

However, enabling language to help vision in this decoupled regime is not straightforward. The two modalities differ in inductive biases, and noise characteristics: Low-level sensor noise for vision (Tian et al., 2020) and semantic and lexicographical ambiguity for language (Abeysiriwardana & Sumanathilaka, 2024; Haber & Poesio, 2024). Naively forcing alignment can distort useful visual invariances, and aggressively coupling objectives risks overfitting to textual idiosyncrasies rather than robust features. Moreover, any auxiliary signal should be lightweight, easy to integrate with common ViT backbones, and ideally removable at inference time.

To address these challenges, we introduce *TextTeacher*, a drop-in auxiliary training objective that injects textual semantics into a vision-only model. To instantiate our approach at scale, we use pretrained captioning models to generate captions for ImageNet, which lacks textual annotations. We find that concise descriptions containing additional information to the label work best. Concretely, *TextTeacher* uses these image captions and extracts high-level semantic embeddings by processing them with a frozen, pretrained text encoder. A lightweight projection aligns the vision backbone's representation space with these embeddings and an auxiliary loss nudges image features toward their corresponding textual anchors. Crucially, the text encoder and projection are used *only during training*; the resulting model remains a standard, pure vision network at inference, incurring no additional latency or parameters.

This design yields three practical benefits: First, language-derived structure shapes the loss landscape and accelerates the emergence of semantically organized features, thus guiding model convergence to a better optimum and reducing overfitting. Second, *TextTeacher* provides additional yet semantically aligned information, mitigating the impact of noisy or ambiguous labels. Third, by relying on frozen text encoders and small adapters, we benefit from the shared representation geometry without the cost and complexity of multimodal training of the target model.

We validate these claims empirically on ImageNet and downstream benchmarks. With the same training recipe and budget, *TextTeacher* boosts accuracy by up to +2.7 p.p. on ImageNet (+8.4 p.p. under noisy labels) and improves transfer performance by +1.0 p.p. on average for transformers on diverse fine-grained tasks, all with negligible overhead. We find that *TextTeacher* is most effective when injecting information that is complementary to both pixels and labels. When comparing to knowledge distillation from unimodal

or multimodal pretrained models, *TextTeacher* is significantly more efficient, outperforming baselines at a fixed compute budget or alternatively reaching the same or better accuracy at only 66% the compute time. Our ablations indicate that *TextTeacher* functions as a feature-space preconditioner in the early stages of training, which acts mostly at deeper model layers.

**Contributions:**

- We pose and investigate a fundamental question: *Can the semantic knowledge of a language model efficiently improve a vision model?* We operationalize this question in a co-learning setting via a decoupled auxiliary signal derived from a frozen text encoder.

- We introduce *TextTeacher*, a general approach that efficiently injects textual knowledge into standard vision backbones using an auxiliary alignment loss. *TextTeacher* avoids multimodal pretraining of the target vision model and requires no text components at inference.

- We provide extensive experiments showing that *TextTeacher* improves accuracy and transfer, especially under label noise, and acts as a feature-space preconditioner that shapes early training dynamics. We see that our language-based auxiliary guidance outperforms visual guidance and knowledge distillation.

Overall, *TextTeacher* offers a low-cost, modality-decoupled and PRH-inspired route for importing semantic priors from language into vision, and narrowing the gap between unimodal training pipelines and the semantic richness typically associated with multimodal training while preserving the simplicity and efficiency of pure vision inference.

## 2 Related Work

**Contrastive Multimodal Pretraining.** CLIP (Radford et al., 2021) popularized dual-encoder contrastive pretraining over web-scale image–text pairs (400M), enabling zero-shot transfer from text. ALIGN (Jia et al., 2021) scales up the dataset size to 1.8 billion noisy pairs. Subsequent work refines the textual signal or augments objectives: A Fistful of Words (Tejankar et al., 2022) uses bag-of-words targets. SLIP (Mu et al., 2022) couples contrastive learning with SimCLR's (Chen et al., 2020) self-supervision. TIPS (kokitsi Maninis et al., 2025) integrates masked image modeling into CLIP-style training, while AVSE (Liu et al., 2025) injects locality via sub-image/sub-text embeddings. In contrast, we forego web-scale pretraining altogether and train ImageNet classifiers directly, using text as an auxiliary signal for classification.

**Captioning as a Training Task.** In another line of work, language generation has been used to imbue visual features with semantic structure. VirTex (Desai & Johnson, 2021) pretrains a ConvNet for captioning before finetuning for recognition tasks. CoCa (Yu et al., 2022) unifies caption generation with a CLIP-like image-text contrastive loss. The task-specific DIC-Transformer (Zeng et al., 2024) adds captioning as an auxilliary task to increase interpretability of plant disease classification. While effective, captioning setups require heavy text decoders and learning to generate. Our approach aligns image features to frozen text embeddings, avoiding a generative head.

**Multimodal Knowledge Distillation.** A complementary direction distills supervision across modalities to compress powerful VLM teachers into compact students. Fang et al. (2021) distill from one VLM to another via on hidden embeddings and attention distributions. VL2Lite (Jang et al., 2025) compresses CLIP into a lightweight image encoder for small-scale image classification task using prompt templates as text input. Guo et al. (2025) replace fixed prompts with learnable text embeddings initialized from WordNet hypernyms. MILAN (Hou & Kung, 2025) pretrains a masked autoencoder while using CLIP image features to weigh patch importance. In comparison, *TextTeacher* differs by dispensing with a multimodal teacher entirely: We rely on a unimodal language model and off-the-shelf captions, reducing compute and the risk of test-image leakage through massive web corpora.

**Text-Guided Image Classification Training.** Closer to our setting, several works add semantically informed signals during classification training. DeViSE (Frome et al., 2013) maps images into a Word2Vec

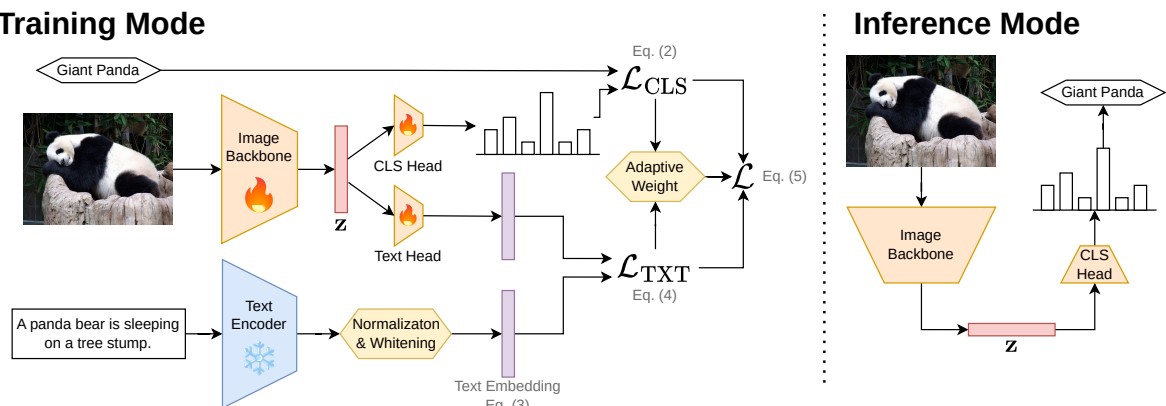

Figure 1: Setup of text-guided image-classification using *TextTeacher*: During training, we use a frozen text encoder on image captions to inject with semantic knowledge into an image classifier. At inference time, we deploy a standard, unimodal vision model.

label space and related methods additionally exploit class attributes (Akata et al., 2015). Feuer et al. (2022) extract classification labels from captions, increasing robustness. Most similar to *TextTeacher*, BorLan (Ma et al., 2023) uses prompt templates to build Gaussian targets in BERT-L's (Devlin et al., 2019) output space to align image embeddings to. In contrast, *TextTeacher* utilizes per-image captions to not only re-encode the label information but include additional instance-level information.

Across prior work, multimodal text supervision consistently shapes strong visual features. We study a complementary point: Direct ImageNet training where captions act only as a lightweight training-time preconditioner, improving semantics without changing test-time cost.

## 3 *TextTeacher*

Our goal is to use a pure text model as a training-time teacher to improve a vanilla vision model, with no text used at inference. We first recap ViT classification, then describe how we extract and normalize language targets, and finally introduce *TextTeacher*: a dual-head training objective with adaptive loss weighting.

### 3.1 Image Classification with Transformers

We adopt a standard ViT (Dosovitskiy et al., 2021) classifier setup that models the image domain as a sequence-encoding problem. First, an input image $\mathbf{x} \in \mathbb{R}^{H \times W \times 3}$ is split into $N = \frac{HW}{P^2}$ non-overlapping patches of size $P \times P$. Each flattened patch is then projected into $d \in \mathbb{N}$ dimensions by a learnable matrix $\mathbf{W}_p \in \mathbb{R}^{3P^2 \times d}$. After prepending a learnable class token [CLS] and adding positional embeddings, we obtain the input sequence $\mathbf{X}_0 \in \mathbb{R}^{(N+1) \times d}$, which is processed by an $L$-layer Transformer encoder $T_{\text{enc}}(\,\cdot\,; \theta)$ with trainable parameters $\theta$. Each encoder layer consists of multi-head self-attention followed by an MLP block. The final image embedding $\mathbf{z}$ is the encoding of the [CLS] token:

$$\mathbf{z} = T_{\text{enc}}(\mathbf{X}_0; \theta)[\text{CLS}] \in \mathbb{R}^d. \tag{1}$$

A linear head $H_{\text{cls}} \in \mathbb{R}^{d \times C}$ produces output probabilities $\mathbf{p}_{\text{cls}} = \text{softmax}(\mathbf{z} H_{\text{cls}})$ over $C$ classes. For classification, we optimize over the standard cross-entropy loss:

$$\mathcal{L}_{\text{cls}} = CE(\mathbf{p}_{\text{cls}}, \, y), \tag{2}$$

where $y$ is the image's ground-truth label. All trainable parameters $(\theta, \mathbf{W}_p, H_{\text{cls}})$ are updated end-to-end using backpropagation.

## 3.2 Text as a Knowledge Source

We treat language as a structured prior over visual concepts. Concretely, each training image is paired with a caption that we embed with a frozen text encoder to obtain a semantic target. We then normalize these targets to expose a well-behaved "knowledge manifold" that regularizes the visual representation during training.

Curating large human-annotated image–text corpora is costly and misaligned with our goal of both lightweight training and having captions for the class-labeled images we need for image classification. Instead, we require only that each labeled training image $\mathbf{x}$ be associated with one caption cap$[\mathbf{x}]$. In practice, we obtain cap$[\mathbf{x}]$ from off-the-shelf captioners (e.g., BLIP-L (Li et al., 2023), CoCa (Yu et al., 2022), Dragonfly (Thapa et al., 2024), PaliGemma (Beyer et al., 2024)) and, when desired, treat caption length as a controlled variable by extracting keyword-style summaries using a language model (Yang et al., 2025). This yields, for every image, both a human-annotated label $y$, which is used by the classifier, and a text description cap$[\mathbf{x}]$ for the language encoder.

We evaluate encoder families with different inductive biases, including the encoder-only BERT (Devlin et al., 2019) and the decoder-based Qwen3-Embedding (Zhang et al., 2025) and NV-Embed (Lee et al., 2025). For completeness, we also test the contrastively image-text pre-trained CLIP text encoder (Radford et al., 2021). Given the caption cap$[\mathbf{x}]$, we obtain the raw $\mathbf{x}$-aligned text embedding $g_t\left(\text{cap}[\mathbf{x}]\right) \in \mathbb{R}^{d_{\text{txt}}}$.

Off-the-shelf embedding spaces often exhibit anisotropy, where variance is concentrated in a few directions, leading to biased similarity structures (Cai et al., 2021; Razzhigaev et al., 2024). To mitigate this, we apply corpus-level whitening. Let $\mu_{g_t}$ and $\Sigma_{g_t}$ be the empirical mean and covariance of $\{g_t(\text{cap}[\mathbf{x}])\}_{\mathbf{x}}$ over the training set. We define the normalized text target $\mathbf{t}$ for an image $\mathbf{x}$ as

$$\mathbf{t} = \Sigma_{g_t}^{-\frac{1}{2}}\left(g_t(\text{cap}[\mathbf{x}]) - \mu_{g_t}\right) \in \mathbb{R}^{d_{\text{txt}}}. \tag{3}$$

This transformation equalizes marginal variances and reduces dominant-direction effects. We conceptualize these embeddings as samples from a semantic manifold $\mathcal{M} \subset \mathbb{R}^{d_{\text{txt}}}$. Points on $\mathcal{M}$ correspond to coherent linguistic concepts like objects, attributes, and relations, while local neighborhoods capture fine-grained semantics. During training, the visual representation $\mathbf{z}$ (Equation (1)) is encouraged to align with $\mathcal{M}$, shaping the geometry of vision features.

We freeze $g_t$, precompute $\mu_{g_t}$ and $\Sigma_{g_t}$, and cache the text embedding $\mathbf{t}$ for all images. This eliminates text-encoder overhead from the training loop and ensures that gradients flow only through the vision model.

## 3.3 Text to Guide Vision Training

After constructing knowledge-rich text embeddings $\mathbf{t} \in \mathcal{M}$, we leverage them to condition the space of vision-embeddings and guide the learning of vision features through a lightweight dual-head design. The setup of *TextTeacher* is visualized in Figure 1. Alongside the classification head $H_{\text{cls}} \in \mathbb{R}^{d \times C}$, we attach a text head $H_{\text{txt}} \in \mathbb{R}^{d \times d_{\text{txt}}}$ to the image representation $\mathbf{z}$ (Equation (1)). Both heads work in tandem to produce two predictions: A classification prediction $\mathbf{p}_{\text{cls}} = \text{softmax}(\mathbf{z}H_{\text{cls}}) \in \mathbb{R}^{C}$ and a text alignment prediction $\mathbf{p}_{\text{txt}} = \mathbf{z}H_{\text{txt}} \in \mathbb{R}^{d_{\text{txt}}}$. We train the classifier with standard cross-entropy on the classification prediction (see Equation (2)) and use an auxiliary CLIP-style contrastive loss to align the predicted text vectors $\mathbf{p}_{\text{txt}}$ with the semantic targets $\mathbf{t}$. The text loss over one batch of size $B \in \mathbb{N}$ is:

$$\begin{aligned}
\mathcal{L}_{\text{txt}} &= \mathcal{L}_{\text{CLIP}}\left([\mathbf{p}_{\text{txt}}^{(1)}, ..., \mathbf{p}_{\text{txt}}^{(B)}], [\mathbf{t}^{(1)}, ..., \mathbf{t}^{(B)}]\right) \\
&= -\frac{1}{B}\sum_{j=1}^{B}\log\frac{\exp\langle\mathbf{p}_{\text{txt}}^{(j)}, \mathbf{t}^{(j)}\rangle}{\sum_{k=1}^{B}\exp\langle\mathbf{p}_{\text{txt}}^{(j)}, \mathbf{t}^{(k)}\rangle} - \frac{1}{B}\sum_{j=1}^{B}\log\frac{\exp\langle\mathbf{p}_{\text{txt}}^{(j)}, \mathbf{t}^{(j)}\rangle}{\sum_{k=1}^{B}\exp\langle\mathbf{p}_{\text{txt}}^{(k)}, \mathbf{t}^{(j)}\rangle}.
\end{aligned} \tag{4}$$

This objective pulls each prediction toward its corresponding text target while repelling it from others. To balance classification and alignment, we form the total loss

$$\mathcal{L} = \lambda_t \alpha_{\text{adapt}} \mathcal{L}_{\text{text}} + (1 - \lambda_t)\mathcal{L}_{\text{cls}}, \tag{5}$$

where $\lambda_t \in [0,1]$ trades off the two terms and $\alpha_{\text{adapt}}$ is an adaptive weight to equalize gradient magnitudes from both losses similar to Yao et al. (2025). For computational reasons, we approximate the fraction of gradient magnitudes at the model weights by the magnitudes at the shared representation $\mathbf{z}$:

$$\alpha_{\text{adapt}} := \frac{\left\| \frac{\partial \mathcal{L}_{\text{cls}}}{\partial \mathbf{z}} \right\|}{\left\| \frac{\partial \mathcal{L}_{\text{txt}}}{\partial \mathbf{z}} \right\|} \sim \frac{\| \nabla_\theta \mathcal{L}_{\text{cls}} \|}{\| \nabla_\theta \mathcal{L}_{\text{txt}} \|}, \tag{6}$$

so that, for a fixed $\lambda_t$, the two contributions have a comparable scale. We consider different $\lambda_t$ schedules over time: (*i*) constant $\lambda_t \equiv \lambda$, (*ii*) linear or cosine annealing, and (*iii*) stepping down from $\lambda$ to 0 (over 10 epochs) at a predefined point during training. All three fit seamlessly into Equation (5).

Intuitively, the text head provides a semantic scaffold: It projects the image features into the semantic manifold $\mathcal{M}$, encouraging neighborhoods in vision-space to respect neighborhoods in text-space. The classifier continues to optimize separability for the task labels. By sharing the same backbone, improvements in local geometry induced by the text-alignment loss can transfer to the classification decision boundary. At inference time, the text head can be discarded (see Figure 1; right).

## 4 Experiments

We evaluate *TextTeacher* on ImageNet and downstream classification, quantify gains across architectures, and compare against other guiding and distillation methods. Full hyperparameter ablations (e.g., $\lambda_t$, knowledge source, text encoder) appear in Sections 4.3 and 4.5. We analyze *TextTeacher*'s effect on model weights in Section 4.6 and use *TextTeacher* to mitigate label noise in Section 4.7.

### 4.1 Experimental Setup

We train our models from scratch on ImageNet (Deng et al., 2009) and report top-1 accuracy on the standard validation set. Following Touvron et al. (2022) and Nauen et al. (2025), we use AdamW with a learning rate of 0.003 with cosine decay and linear warmup and a batch size of 2048 for 100 or 300 epochs. DeiT uses the data augmentation and hyperparameters of Touvron et al. (2021). Experiments are performed on 4 NVIDIA A100s/H100s. For details see Section A. Unless specified, we extract per-image tags using Qwen3 from CoCa captions and encode them with a frozen BERT-Large text encoder. We always report mean $\pm$ standard deviation over 3 independent seeds.

### 4.2 Image Classification Results

This experiment tests whether *TextTeacher* improves accuracy for different model architectures. Table 1 shows consistent gains across ViT (Dosovitskiy et al., 2021) and DeiT (Touvron et al., 2021) with patch size 16, Swin (Liu et al., 2021) with patch size 4 and window size 7, Next-ViT (Li et al., 2022), XCiT (El-Nouby et al., 2021) and ResNet (He et al., 2016) when using a constant $\lambda_t \equiv 0.5$ for 300 epochs. *TextTeacher* consistently boosts accuracy across these 15 models; typically between +0.6 p.p. and +1.5 p.p. and up to +2.7 p.p. for DeiT-L. These results highlight the architecture-agnostic benefit of integrating *TextTeacher* during training, demonstrating robust enhancements over strong baselines.

We ask whether preconditioning with *TextTeacher* on ImageNet transfers to fine-grained recognition without using text at finetuning as the representation space has already been conditioned in pretraining. We finetune on FGVC-Aircraft (Maji et al., 2013), Caltech-UCSD Birds (Wah et al., 2011) Stanford Cars (Krause et al., 2013), Oxford Flowers (Nilsback & Zisserman, 2008), Food-101 (Kaur et al., 2017), and Oxford-IIIT Pets (Parkhi et al., 2012). Table 3 shows that gains on ImageNet carry over for transformer backbones (mean +0.9 p.p.), while ResNets exhibit smaller or occasionally negative changes depending on their size. We hypothesize that because of their larger capacity (Chen et al., 2022) and `[CLS]`-token, a natural anchor to align to text, transformers benefit more from the additional guiding signal, while ResNets need to adjust their pixel representations.

Table 2 presents a compute-matched comparison of *TextTeacher* to various baselines grouped by how the auxiliary signal is used. Note that we only include model training compute (in time per epoch) but not

Table 1: ImageNet results with and without *TextTeacher* for different architectures. Constant $\lambda_t \equiv 0.5$ for 300 epochs. *TextTeacher* shows consistent improvements for all models.

| Model | Baseline | $+\textit{TextTeacher}$ | Delta |
|---|---|---|---|
| ViT-S | $79.1 \pm 0.1$ | $80.5 \pm 0.2$ | +1.4 |
| ViT-B | $77.6 \pm 0.2$ | $79.3 \pm 0.4$ | +1.7 |
| ViT-L | $75.3 \pm 0.4$ | $76.8 \pm 0.2$ | +1.5 |
| DeiT-S | $80.1 \pm 0.1$ | $80.2 \pm 0.2$ | +0.1 |
| DeiT-B | $81.9 \pm 0.3$ | $82.3 \pm 0.2$ | +0.4 |
| DeiT-L | $79.3 \pm 2.3$ | $82.0 \pm 0.6$ | +2.7 |
| Swin-Ti | $77.9 \pm 0.2$ | $79.3 \pm 0.1$ | +1.4 |
| Swin-S | $79.4 \pm 0.1$ | $80.6 \pm 0.1$ | +1.2 |
| XCiT-Ti | $80.6 \pm 0.1$ | $80.8 \pm 0.1$ | +0.2 |
| XCiT-S | $80.8 \pm 0.1$ | $81.6 \pm 0.1$ | +0.8 |
| XCiT-M | $78.8 \pm 0.1$ | $79.8 \pm 0.1$ | +1.0 |
| NextViT-S | $81.2 \pm 0.1$ | $81.8 \pm 0.1$ | +0.6 |
| NextViT-B | $81.5 \pm 0.1$ | $82.1 \pm 0.1$ | +0.6 |
| ResNet50 | $78.3 \pm 0.1$ | $79.1 \pm 0.1$ | +0.8 |
| ResNet101 | $79.4 \pm 0.1$ | $80.4 \pm 0.1$ | +1.0 |

Table 2: ImageNet accuracy (100 epochs) of *TextTeacher* compared to various methods. We report the training time for ViT-B in GPU-minutes on 4 NVIDIA H100s. **Bold**: best per block. Underlined: best overall. For Implementation details see Section B.

| Method | Accuracy [%] with | | $\frac{\text{time}}{\text{epoch}}$ [min] |
|---|---|---|---|
| | ViT-S | ViT-B | ViT-B |
| Classification Only | | | |
| Baseline | $77.6 \pm 0.2$ | $76.5 \pm 0.4$ | $30.7 \pm 2.3$ |
| Knowledge distillation (online; full training; $\approx 150\%$ compute) | | | |
| CLIP-ViT-L | $77.6 \pm 0.2$ | $78.6 \pm 0.1$ | $48.2 \pm 1.0$ |
| CoCa | $77.8 \pm 0.1$ | $78.7 \pm 0.1$ | $50.3 \pm 6.8$ |
| DINOv2-L | $\mathbf{78.4} \pm 0.1$ | $\mathbf{79.1} \pm 0.3$ | $47.8 \pm 0.4$ |
| VL2Lite | $75.7 \pm 0.2$ | $76.2 \pm 0.4$ | $47.8 \pm 0.4$ |
| Knowledge distillation (online; compute matched) | | | |
| CLIP-ViT-L | $73.8 \pm 0.2$ | $78.1 \pm 0.1$ | $48.2 \pm 1.0$ |
| CoCa | $73.5 \pm 0.4$ | $77.6 \pm 0.3$ | $50.3 \pm 6.8$ |
| DINOv2-L | $\mathbf{74.3} \pm 0.2$ | $\mathbf{78.9} \pm 0.2$ | $47.8 \pm 0.4$ |
| VL2Lite | $68.4 \pm 0.4$ | $75.2 \pm 0.3$ | $47.8 \pm 0.4$ |
| Vision Guided (offline) | | | |
| CLIP-ViT-B | $77.9 \pm 0.1$ | $78.6 \pm 0.2$ | $33.1 \pm 0.2$ |
| CLIP-ViT-L | $77.8 \pm 0.2$ | $\mathbf{78.8} \pm 0.2$ | $32.8 \pm 1.2$ |
| CoCa | $\mathbf{78.0} \pm 0.1$ | $78.5 \pm 0.1$ | $31.9 \pm 3.0$ |
| DINOv2-B | $77.9 \pm 0.2$ | $78.4 \pm 0.4$ | $30.6 \pm 2.8$ |
| DINOv2-L | $77.8 \pm 0.1$ | $78.6 \pm 0.6$ | $31.2 \pm 2.9$ |
| Text Guided (offline) | | | |
| VL2Lite (text only) | $77.0 \pm 0.5$ | $75.5 \pm 0.3$ | $31.4 \pm 2.5$ |
| BorLan (distribution) | $76.7 \pm 0.5$ | $79.0 \pm 0.4$ | $32.9 \pm 0.1$ |
| BorLan + adaptive weight | $77.7 \pm 0.3$ | $78.2 \pm 0.1$ | $31.1 \pm 2.5$ |
| *TextTeacher* (sample) | $\underline{\mathbf{78.4}} \pm 0.1$ | $\underline{\mathbf{79.1}} \pm 0.6$ | $32.1 \pm 1.6$ |

preprocessing compute in Table 2. Offline guidance methods are all trained for 100 epochs, while online knowledge distillation in the compute-matched setting is trained for $\frac{2}{3} \times 100$ epochs, since the time per epoch is 50% higher on an H100 for ViT-B. *TextTeacher* outperforms all methods that use an auxiliary offline text- or vision-based signal with image classification for both ViT-S and ViT-B. Online distillation methods are less efficient due to running the teacher model alongside the student during training. Thus, in a compute-matched setting, they underperform text guided methods like BorLan and *TextTeacher*. When utilizing the full training time, online distillation with DINOv2-L becomes our strongest baseline, which reaches the same performance as *TextTeacher*, but is $\approx 50\%$ slower. Even when adding the preprocessing compute (captioning and embedding the captions) for *TextTeacher* to the runtime, it is still faster by $\approx 6$ GPU-hours when training ViT-B for 300 epochs compared to online knowledge distillation. These results underscore that our sample-level language signals provide more effective and compute-efficient supervisory signals than distribution-level or even sample-level vision-based guidance or knowledge distillation.

### 4.3 How to Handle Text for Vision

*What textual signal and representation best guide a vision backbone during training with TextTeacher?* In this section, we ablate the knowledge source and text embedding model used for *TextTeacher*. We utilize three sources of textual knowledge with varying granularity: (*i*) Labels, (*ii*) synthetic captions, (*iii*) tags extracted from these captions, representing a shortened, condensed form of the captions, and (*iv*) a naive mix of labels and captions. Table 4 shows that Qwen3's Tags from CoCa captions (CoCa→Qwen3 Tags) yield the best accuracy (78.4%) despite being the shortest text (3.30 words on average). Generally, extracting tags from captions slightly increases the accuracy compared to the original captions, suggesting that *TextTeacher* benefits from *attribute-dense* encoded tags that align with decision-relevant directions, rather than verbose descriptions. However, this gain comes at the cost of additional preprocessing with a large language model. For practical deployments, especially when training only a few models, we recommend using captions directly to minimize preprocessing overhead. Notably, a naive label+caption mix (77.9%) underperforms just the

Table 3: Downstream finetuning results of models from Table 1. The *TextTeacher*-column indicates if *TextTeacher* has been used during ImageNet *pretraining*. No text is used during finetuning.

| Model | *TextTeacher* | Aircraft | Birds | Cars | Flowers | Food | Pets | Mean |
|---|---|---|---|---|---|---|---|---|
| ViT-S | ✗ | $72.4 \pm 1.0$ | $78.4 \pm 0.6$ | $89.8 \pm 0.3$ | $94.5 \pm 0.2$ | $89.1 \pm 0.1$ | $93.8 \pm 0.2$ | |
| | ✓ | $74.2 \pm 0.3$ | $79.4 \pm 0.1$ | $90.4 \pm 0.2$ | $95.1 \pm 0.3$ | $89.3 \pm 0.1$ | $94.1 \pm 0.1$ | |
| | Delta | +1.8 | +1.0 | +0.6 | +0.6 | +0.2 | +0.3 | +0.8 |
| ViT-B | ✗ | $71.7 \pm 0.5$ | $78.2 \pm 0.7$ | $90.0 \pm 0.2$ | $94.8 \pm 0.4$ | $89.8 \pm 0.2$ | $94.1 \pm 0.4$ | |
| | ✓ | $73.3 \pm 0.5$ | $80.2 \pm 0.3$ | $91.1 \pm 0.3$ | $95.6 \pm 0.4$ | $90.3 \pm 0.2$ | $94.3 \pm 0.2$ | |
| | Delta | +1.6 | +2.0 | +1.1 | +0.8 | +0.5 | +0.2 | +1.0 |
| ViT-L | ✗ | $72.1 \pm 1.0$ | $78.3 \pm 0.2$ | $88.8 \pm 0.3$ | $94.4 \pm 0.3$ | $90.1 \pm 0.2$ | $94.2 \pm 0.4$ | |
| | ✓ | $72.2 \pm 0.9$ | $80.4 \pm 1.4$ | $89.9 \pm 0.5$ | $95.3 \pm 0.4$ | $90.5 \pm 0.2$ | $94.8 \pm 0.3$ | |
| | Delta | +0.1 | +2.1 | +1.1 | +0.9 | +0.4 | +0.6 | +0.9 |
| Swin-Ti | ✗ | $77.0 \pm 0.1$ | $81.5 \pm 0.4$ | $91.3 \pm 0.6$ | $95.9 \pm 0.1$ | $90.0 \pm 0.2$ | $94.2 \pm 0.1$ | |
| | ✓ | $79.6 \pm 0.9$ | $82.8 \pm 0.9$ | $92.1 \pm 0.4$ | $96.0 \pm 0.1$ | $90.4 \pm 0.2$ | $94.1 \pm 0.1$ | |
| | Delta | +2.6 | +1.3 | +0.8 | +0.1 | +0.4 | -0.1 | +0.9 |
| Swin-S | ✗ | $75.7 \pm 1.4$ | $82.9 \pm 0.8$ | $91.0 \pm 0.3$ | $95.9 \pm 0.5$ | $91.1 \pm 0.2$ | $94.4 \pm 0.1$ | |
| | ✓ | $78.5 \pm 0.8$ | $83.8 \pm 0.5$ | $92.2 \pm 0.3$ | $96.2 \pm 0.1$ | $91.2 \pm 0.1$ | $94.5 \pm 0.2$ | |
| | Delta | +2.8 | +0.9 | +1.2 | +0.3 | +0.1 | +0.1 | +0.9 |
| ResNet50 | ✗ | $78.2 \pm 0.5$ | $79.5 \pm 0.1$ | $89.8 \pm 0.2$ | $91.7 \pm 0.4$ | $84.4 \pm 0.2$ | $93.7 \pm 0.3$ | |
| | ✓ | $78.0 \pm 0.3$ | $79.3 \pm 0.4$ | $89.1 \pm 0.3$ | $90.9 \pm 0.2$ | $84.3 \pm 0.1$ | $93.3 \pm 0.3$ | |
| | Delta | -0.2 | -0.2 | -0.7 | -0.8 | -0.1 | -0.2 | -0.4 |
| ResNet101 | ✗ | $78.4 \pm 0.6$ | $79.8 \pm 0.2$ | $90.3 \pm 0.1$ | $91.2 \pm 0.5$ | $86.0 \pm 0.2$ | $94.3 \pm 0.2$ | |
| | ✓ | $78.7 \pm 0.6$ | $79.9 \pm 0.1$ | $90.1 \pm 0.1$ | $91.2 \pm 0.4$ | $86.1 \pm 0.1$ | $94.1 \pm 0.2$ | |
| | Delta | +0.3 | +0.1 | -0.2 | $\pm 0.0$ | +0.1 | -0.2 | $\pm 0.0$ |

Table 4: Knowledge source ablation of ViT-S for 100 epochs on ImageNet with $\lambda_t \equiv 0.5$ and BERT-L text encoder. Short, attribute-dense tags extracted from strong captioners perform best.

| Knowledge Source | Mean Length [w] | Synth. | Accuracy [%] |
|---|---|---|---|
| Dragonfly | 81.87 | ✓ | $77.8 \pm 0.3$ |
| PaliGemma | 31.94 | ✓ | $78.0 \pm 0.1$ |
| CoCa | 12.58 | ✓ | $78.3 \pm 0.1$ |
| BLIP-L | 11.64 | ✓ | $78.1 \pm 0.4$ |
| Dragonfly→Qwen3 Tags | 5.69 | ✓ | $78.0 \pm 0.3$ |
| CoCa→Qwen3 Tags | 3.30 | ✓ | $78.4 \pm 0.1$ |
| Labels + CoCa | 14.31 | ∼ | $77.9 \pm 0.1$ |
| Labels | 5.50 | ✗ | $77.4 \pm 0.2$ |

Table 5: Text encoder ablation for ViT-S on ImageNet for 100 epochs with CoCa→Qwen3 Tags. Encoders are closely matched when properly whitened. Removing whitening hurts both mean accuracy and stability for BERT and CLIP.

| Embedding Model | $d_{embed}$ | Whitening | Accuracy [%] |
|---|---|---|---|
| BERT-B | 768 | ✓ | $78.4 \pm 0.1$ |
| BERT-L | 1024 | ✓ | $78.4 \pm 0.1$ |
| BERT-L | 1024 | ✗ | $77.8 \pm 0.5$ |
| Qwen3-Embedding | 4096 | ✓ | $77.8 \pm 0.3$ |
| Qwen3-Embedding | 4096 | ✗ | $78.0 \pm 0.1$ |
| NV-Embed | 4096 | ✓ | $77.4 \pm 0.1$ |
| NV-Embed | 4096 | ✗ | $77.9 \pm 0.4$ |
| CLIP-Text-B | 512 | ✓ | $78.1 \pm 0.1$ |
| CLIP-Text-L | 768 | ✓ | $78.3 \pm 0.1$ |
| CLIP-Text-L | 768 | ✗ | $78.0 \pm 0.5$ |

captions (78.3%). Just the labels alone (77.4%) do not outperform the text-free baseline (77.5%), indicating that *TextTeacher* works best when providing additional information to the vision model. Fixing our knowledge source to CoCa→Qwen3 Tags, we investigate different embedding models in Table 5. Surprisingly, most encoders are very close in performance when used with their best whitening configuration and even the already image-aligned CLIP-Text does not outperform the unimodal models. Whitening improves both mean and variance for BERT and CLIP, consistent with reducing embedding anisotropy, leading to a more robust alignment target.

Interestingly, while the knowledge source has a larger impact on *TextTeacher*, the text embedding model does not make much of a difference. We adopt CoCa→Qwen3 Tags encoded by BERT-L with whitening as our default setting for all other experiments.

Table 6: Classifying images based on their text encoding. Accuracy is lower than when using *TextTeacher* and, importantly, is not improved by using whitening.

| Captions | Encoder | Mean Normalize | Std. Normalize | Accuracy |
|---|---|---|---|---|
| CoCa | BERT-B | ✗ | ✗ | 52.8 |
| CoCa | BERT-B | ✗ | mean | 52.6 |
| CoCa | BERT-B | ✗ | full | 52.5 |
| CoCa | BERT-B | ✓ | ✗ | 53.1 |
| CoCa | BERT-B | ✓ | mean | 52.5 |
| CoCa | BERT-B | ✓ | full | 52.4 |

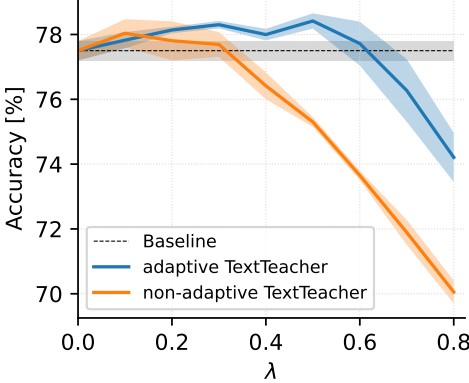

Figure 2: $\lambda$ sweep for ViT-S on ImageNet (100 epochs) with $\lambda_t \equiv \lambda \in [0.0, 0.8]$. *TextTeacher* improves over the baseline especially at $\lambda = 0.5$ with $\alpha_{\text{adapt}}$. Without adaption, accuracy drops sharply for $\lambda > 0.3$; with adaption it is stable up to $\lambda = 0.6$.

Table 7: Schedule ablation at fixed peak $\lambda = 0.5$ (ViT-S, 100 epochs, $\alpha_{\text{adapt}}$ enabled). Schedules are ordered by their early-time mass (value $\lambda_\epsilon$ at small $\epsilon > 0$). Higher early mass correlates with higher accuracy.

| Schedule | | $\lambda$ | Accuracy [%] | Delta |
|---|---|---|---|---|
| – | | 0.0 | $77.5 \pm 0.3$ | |
| const | ⊏ | 0.5 | $78.4 \pm 0.1$ | +0.9 |
| halfcos | ◹ | 0.5 | $78.3 \pm 0.1$ | +0.8 |
| cos | ◹ | 0.5 | $78.2 \pm 0.1$ | +0.7 |
| linear | ◹ | 0.5 | $77.9 \pm 0.3$ | +0.4 |

### 4.4 Text Encoders as Image Classifiers

To better understand the contribution of the language branch in our training-only pipeline, we evaluate how much image-classification signal is recoverable from captions alone. This experiment isolates the text side of the system: We use CoCa captions encoded with BERT-B, and train a single linear layer as a classifier on top of the resulting embeddings. As shown in Table 6, the resulting accuracies cluster tightly around 52–53%, with mean-only normalization yielding the strongest performance (53.1%), and full whitening providing no consistent benefit. This is in contrast to the full *TextTeacher* setup, where whitening provides a noticeable accuracy boost when using BERT (see Table 5). These results indicate that while captions contain moderately informative cues about object identity, they lack the granularity required for high-accuracy recognition, reinforcing that our method relies on alignment rather than blindly adhering to text-based signals.

### 4.5 $\lambda$ and $\lambda_t$-Schedules

We study the guidance weight $\lambda_t$ in two parts: First, a scalar sweep with $\lambda_t \equiv \lambda \in [0.0, 0.8]$ with and without adaptive weight $\alpha_{\text{adapt}}$ (Equation (6)); and second, schedule shapes at a fixed peak $\lambda = 0.5$. For this experiment, we train ViT-S on ImageNet for 100 epochs. Figure 2 shows that *TextTeacher* is effective at small $\lambda$ even without adaptation, but becomes brittle as $\lambda$ grows. Adaptation not only increases accuracy, but also widens the stable range to $\lambda = 0.6$ with a peak at $\lambda = 0.5$.

At this fixed peak value of $\lambda = 0.5$, Table 7 shows that schedule shapes that retain higher value in early training (*const*, *half-cos*) perform best, while the *linear* schedule that quickly reduces guidance underperforms.

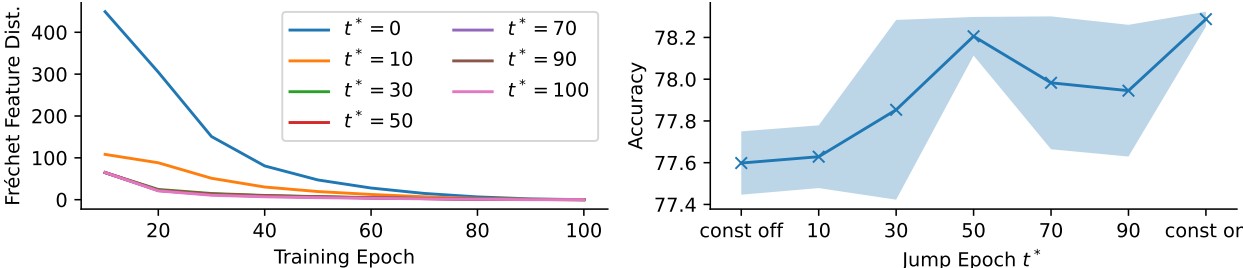

Figure 3: Jump schedules for ViT-S on ImageNet. At epoch $t^*$, $\lambda_t$ decays linearly from 0.5 to 0.0 over 10 epochs. Thus separating the effects of *TextTeacher* over the course of training. **Left:** Fréchet feature distance (FFD) at current training epoch to the final trained model. For $t^* > 30$ the FFD plots coincide. **Right:** Final ImageNet accuracy when jumping at $t^*$. *TextTeacher* accelerates convergence of the feature representation. It reduces the FFD to the final training state early on from 450 to 65 at epoch 10. These gains saturate at epoch $t^* = 50$, while late drops ($t^* \in [70, 90]$) harm stability.

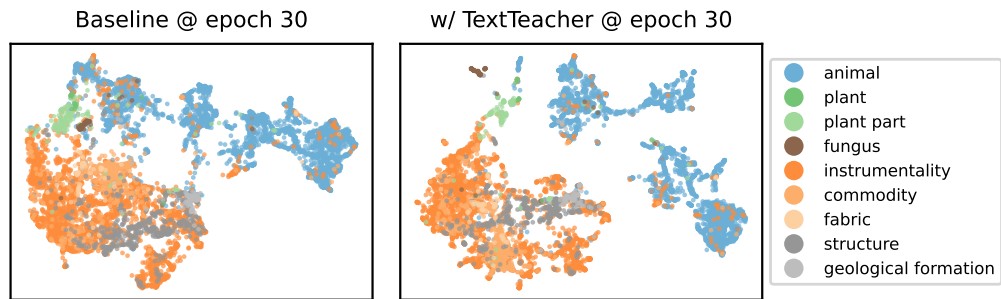

Figure 4: 2D-umap of the embedding space of ViT-S after training without and with *TextTeacher* for 30 (out of 100) epochs. We plot embeddings of 10 000 random validation images with colors marking high-level hypernyms of image classes. Utilizing *TextTeacher* leads to a clearer separability early on in training, especially between the two large groups of *animal* and *instrumentality* and for the clusters of *plant* and *fungus*.

This pattern supports the view of *TextTeacher* as an early-phase regularizer. For all main experiments, we thus choose the constant schedule $\lambda_t \equiv 0.5$ with $\alpha_{\text{adapt}}$.

## 4.6 Effect on Model Weights

*How does TextTeacher affect a model during training?* We begin by analyzing the time effect in training by employing *jump schedules* that hold $\lambda_t = 0.5$ and then drop it linearly to 0 over 10 epochs at a chosen epoch $t^*$. Additionally, we quantify the embedding similarity by computing the per-class Fréchet feature distance (FFD) between the embeddings $\mathbf{z}$ of the current model and the final model at epoch 100. The jump-schedule study in Figure 3 reveals a pronounced early-phase effect. When *TextTeacher* is active during the first 10 epochs, the representation is already much closer to its final semantic configuration (FFD 65 at epoch 10) than the baseline without guidance (FFD 450 at the same epoch), and the gap persists as training proceeds (left in Figure 3). These results are very stable, such that the standard deviation range is too small to be visible in Figure 3 (left). Extending guidance through the first 30–50 epochs largely saturates the benefit. When dropping $\lambda_t$ at epoch $t^* = 50$, the accuracy is very close to the best setting: constant $\lambda_t \equiv 0.5$. In contrast, dropping $\lambda_t$ very late ($t^* \in [70, 90]$) degrades both the mean and stability of final accuracy (right in Figure 3), plausibly because the objective switch leaves insufficient time for the classifier to readapt to pure label supervision. We hypothesize that while the main benefit of *TextTeacher* is realized early on, removing the additional objective introduces noise by making the classifier adapt to a new objective function. This can be seen in the jump at epoch $t^* = 50$, which shows high accuracy but larger variance than the constant $\lambda_t \equiv 0.5$ schedule (0.1 p.p. vs. 0.03 p.p.). We qualitatively validate these findings in Figure 4 where we find that *TextTeacher* conditions the model towards class separability even early on in training (at epoch 30).

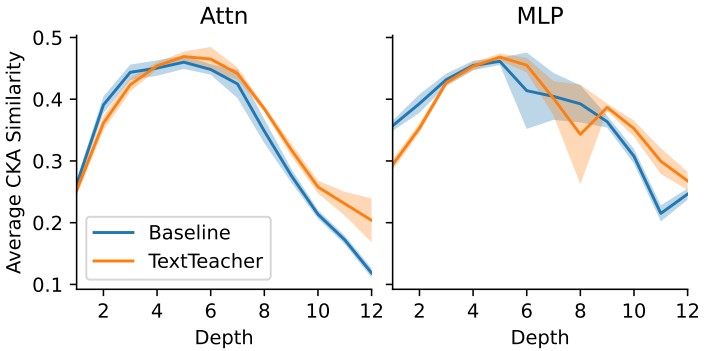

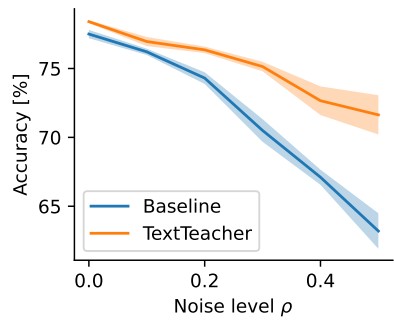

Figure 5: Layer-wise CKA similarity over independent training runs at attention and MLP layers. With *TextTeacher* deeper layers exhibit higher similarity, especially at attention layers.

Figure 6: Label-noise robustness for ViT-S. Accuracy declines with higher noise level $\rho$ for both methods, but *TextTeacher* enlarges its margin over the baseline as $\rho$ increases.

To locate *TextTeacher*'s impact in a trained model, we study how guidance changes representational geometry across random initializations using centered kernel alignment (Kornblith et al., 2019) (CKA) at different depths. Visualizing the layerwise trends, Figure 5 shows that *TextTeacher* increases similarity among deeper representations, while slightly decreasing similarity at earlier layers (direct CKA visualization in Section D.2). This effect is stronger at the output of attention, compared to the output of the MLP layers. Thus, *TextTeacher* primarily organizes higher-level semantic subspaces, while allowing lower-level feature extractors to remain flexible. Together, these results imply that *TextTeacher* is an early-phase, depth-localized regularizer that accelerates convergence toward the shared semantic manifold $\mathcal{M}$ across random seeds.

### 4.7 Robustness under Label Noise

As *TextTeacher* provides an additional training signal, it can inject additional information to counteract noisy labels. We train ViT-S on ImageNet while randomly perturbing labels to a uniform random incorrect class with probability $\rho \in [0.0, 0.5]$. Figure 6 shows that accuracy declines monotonically with $\rho$ for both methods, but the margin of *TextTeacher* widens with higher corruption. At $\rho = 0.5$ the baseline obtains only $63.2 \pm 1.3\%$ while *TextTeacher* reaches $71.6 \pm 1.4\%$, an improvement of 8.4 p.p. Thus, *TextTeacher* stabilizes training and significantly reduces the adverse impact of even high levels of corrupted labels.

## 5 Limitations

While *TextTeacher* assists in classification training, it also has several limitations: First, our preconditioning benefits only realize when training a model from scratch. *TextTeacher* does not assist or may actively harm a model that is finetuned from pretrained (already conditioned) weights by either adding no sizable benefit if the pretrained state aligns with our language guidance or actively harming convergence by pushing towards another non-aligned conditioning (see Section C.3). Second, our method depends on the existence of high-quality, reliable image captions. While neural captioning models continue to improve, they might still reflect biases or may not generalize well to specific domains, like medical images or earth observation. These problems would then be passed on to a model trained with *TextTeacher* on this data. While we hypothesize that concise, human annotated captions would be optimal, we can still profit from ongoing captioner and text-encoder development. And third, *TextTeacher* requires the one-time computational overhead of generating captions for new images if none are available. Captioning the full ImageNet training set (1.3M images) with CoCa (Yu et al., 2022) takes 69 hours on a single NVIDIA H100, which for this paper comes out to 26 minutes per training run with *TextTeacher* where these captions are used – or less than the time for training ViT-B for one more epoch. Tag extraction with Qwen3-32B processes approximately 20 tags/min, requiring $\approx 1000$ GPU-hours for the full ImageNet training set. However, this step is optional and provides only marginal improvements (see Table 4), serving primarily as an analytical tool to understand

effective text representations. To assist practitioners, we release our precomputed captions and embeddings[1]. Future work could explore lightweight extraction methods using smaller language models or rule-based approaches to reduce this preprocessing cost.

## 6 Conclusion

This paper revisits a fundamental question: *Can the semantic knowledge of a language model efficiently improve vision training?* We answer this through *TextTeacher*, a minimal, training-only mechanism that injects textual semantics into standard vision backbones. By using a frozen text encoder to produce semantic anchors and an auxiliary alignment loss to nudge image features, *TextTeacher* improves accuracy and transfer on ImageNet and downstream tasks while keeping the deployed model purely visual. We show that *TextTeacher* acts as a feature-space preconditioner with effects concentrated early in training and at deeper model layers. By utilizing textual cues, *TextTeacher* shapes the loss landscape to guide image embeddings towards a semantic alignment optimum, thus producing more accurate models and preventing overfitting. Looking forward, promising directions include domain-aware text distillation for specialized vocabularies, adaptive schedules that detect when preconditioning has saturated, and extensions to detection and segmentation, where dense regional text could shape fine-grained visual representations. In short, *TextTeacher* offers a lightweight path to import linguistic priors into visual learning: A small change to training that leaves a fast, unimodal model at test time.

### Broader Impact Statement

As already discussed in Section 5, captions generated by neural models trained on web data may contain gender, racial, or cultural biases (Birhane et al., 2021), which could propagate through text embeddings into the vision model's representation space. This indirect transfer from captioner training data, through captions, into visual features may be difficult to detect and audit. Additionally, in specialized domains such as medical imaging or earth observation, general-purpose captioning models may produce inaccurate descriptions, potentially introducing systematic errors. We caution against applying *TextTeacher* in safety-critical settings without domain-appropriate captions and thorough downstream validation across relevant subgroups. We also encourage practitioners to inspect generated captions for systematic biases in sensitive applications and evaluate trained models for fairness. As captioning models improve and are increasingly audited for bias, *TextTeacher* would directly benefit from these advances.

### Acknowledgments

This work was funded by the Carl-Zeiss Foundation under the Sustainable Embedded AI project (P2021-02-009) and by the BMFTR project Albatross (funding code 16IW24002). All compute was done thanks to the Pegasus cluster at DFKI Kaiserslautern. We also thank the anonymous reviewers for their constructive feedback, which helped us improve this paper.

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

## A    Training Hyperparameters

Table 8: Training setup and hyperparameters for our ImageNet training.

| Parameter | ViT & others | DeiT |
|---|---|---|
| Image Resolution | $224 \times 224$ | $224 \times 224$ |
| Epochs | 300 | 300 |
| Learning Rate | 3e-3 | S/B: 1e-3, L: 5e-4 |
| Learning Rate Schedule | cosine decay | cosine decay |
| Batch Size | 2048 | 1024 |
| GPUs | $4\times$ NVIDIA A100/H100/H200 | $4\times$ NVIDIA A100/H100/H200 |
| Warmup Schedule | linear | linear |
| Warmup Epochs | 3 | 3 |
| Weight Decay | 0.02 | 0.05 |
| Label Smoothing | 0.1 | 0.1 |
| Optimizer | Lamb (You et al., 2020) | AdamW |
| Data Augmentation Policy | **3-Augment** (Touvron et al., 2022) | **DeiT** (Touvron et al., 2021) |
| Augmentations | Resize RandomCrop HorizontalFlip Grayscale Solarize GaussianBlur ColorJitter CutMix (Yun et al., 2019) | RandomResizedCrop HorizontalFlip RandomErase (Zhong et al., 2020) RandAugment (Cubuk et al., 2019) ColorJitter Mixup (Zhang et al., 2018) CutMix (Yun et al., 2019) |

Table 9: Training setup for finetuning on different downstream datasets. Other settings are the same as in our ImageNet setup. For finetuning, we always utilize 3-Augment and the related parameters from the *ViT & others* column of Table 8.

| Dataset | Training Images | Batch Size | Epochs | Learning Rate | Num. GPUs |
|---|---|---|---|---|---|
| Aircraft | 3334 | 512 | 500 | 3e-4 | 2 |
| Birds | 5994 | 512 | 500 | 3e-4 | 2 |
| Cars | 8144 | 1024 | 500 | 3e-4 | 4 |
| Flowers | 1020 | 256 | 500 | 3e-4 | 1 |
| Food | 75 750 | 2048 | 100 | 3e-4 | 4 |
| Pets | 3680 | 512 | 500 | 3e-4 | 2 |

For completeness and reproducibility, we list all hyperparameters used in our ImageNet training and downstream evaluations. Unless otherwise noted, models follow the standard ViT (from Touvron et al. (2022); Nauen et al. (2025)) or DeiT (from Touvron et al. (2021)) configurations summarized in Table 8 for ImageNet training and Table 9 listing the differences on downstream datasets. We include optimizer settings, augmentation policies, hardware resources, and dataset-specific finetuning parameters.

## B    Baseline Implementation Details

To ensure a fair and controlled comparison, we re-implement all baselines within the same training framework and architectural template used for our method. This allows us to isolate the effect of each baseline's learning signal without confounding differences in optimization, augmentations, or backbone configuration.

Importantly, all baseline methods are implemented faithfully; we do not modify their formulations. The only differences from the original publications are the training scale and backbone: we train on ImageNet rather

Table 10: Ablation of CLIP distillation setups. We copare our setup to CLIP-Embed-KD (Nair, 2024) and a setting using MSE loss as suggested by Yang et al. (2024).

| Teacher | Adaptive | Loss | ViT-S Accuracy |
|---------|----------|------|----------------|
| Online CLIP-L | ✗ | CLIP | $77.7 \pm 0.1$ |
| Online CLIP-L | ✓ | CLIP | $77.6 \pm 0.2$ |
| Online CLIP-L | ✓ | MSE | $77.7 \pm 0.2$ |

than on the smaller fine-grained datasets used in the original papers, and we use standard ViT backbones. All general training hyperparameters (learning rate, schedule, augmentations, etc.) follow the DeiT/DeiT-III recipes (Touvron et al., 2021; 2022; Nauen et al., 2025), while all method-specific hyperparameters, including loss weights $\lambda$ and schedules, are taken directly from the respective original papers.

### B.1 Knowledge Distillation Baselines

For knowledge-distillation baselines from CLIP and DINOv2, we adopt both offline guidance and online distillation, inserting the teacher supervision into the auxiliary path of our setup so that the additional signal is injected at the same interface point as in *TextTeacher*: **Vision guidance** (offline) precomputes embeddings for each training image using the frozen teacher (CLIP or DINOv2) with minimal augmentation and caches them before training begins. These fixed embeddings are then used in place of *TextTeacher*'s text embeddings during training, keeping the per-epoch cost identical to *TextTeacher*. **Knowledge distillation** (online) computes teacher embeddings on-the-fly by forward-passing the same augmented image views that the student receives through the frozen teacher network. This follows standard knowledge-distillation practice, where the teacher provides view-specific guidance, but incurs approximately 50% additional per-epoch compute cost on an H100 for ViT-B, which we account for in our compute-matched comparison (Table 1). These setups closely follow CLIP-Embed-KD (Nair, 2024), but our setup also adopts our adaptive weighting scheme (Equation (6)). Table 10 compares our setting to CLIP-Embed-KD and to using MSE loss, as suggested in CLIP-KD (Yang et al., 2024). The consistency across loss functions and weighting strategies suggests our vision KD baseline is not disadvantaged by the choice of distillation mechanism.

### B.2 Text-guided Baselines

For VL2Lite, the original formulation projects text embeddings into the image-feature dimension, whereas our framework projects image features into the text-embedding dimension. These two implementations are mathematically equivalent after transposing the projection matrix, since the bilinear inner product is invariant to which side carries the projection:

$$\langle e_{\text{img}} \times W, e_{\text{txt}} \rangle = (e_{\text{img}} \times W) \times e_{\text{txt}}^{\top} = e_{\text{img}} \times (e_{\text{txt}} \times W^{\top})^{\top} = \langle e_{\text{img}}, e_{\text{txt}} \times W^{\top} \rangle .$$

For BorLan, whose formulation does not rely on explicit text embeddings but instead produces an auxiliary class-probability distribution derived from language supervision, we integrate its loss in its original form and attach the predicted distribution to the classification head in our framework. Since BorLan and VL2Lite were not originally evaluated on ImageNet but only on smaller fine-grained datasets, standardizing their implementations within our pipeline ensures that any differences in performance stem from the training signal itself rather than from different experimental setups.

## C  Additional Experiments

### C.1  Comparison to BorLan Under Different Augmentation Strength

We further compare our method to BorLan (Ma et al., 2023) under different augmentation regimes to understand how robustness and performance scale with increasingly strong visual perturbations. This analysis

Table 11: ImageNet comparison of *TextTeacher* and BorLan under increasing augmentation strength. *Mean Delta* is the average improvement over the previous line.

| Method | Accuracy with Augmentation | | | Mean Delta |
|---|---|---|---|---|
| | BorLan | BorLan + CutMix | 3-augment | |
| BorLan (distribution) | $68.3 \pm 0.4$ | $75.3 \pm 0.6$ | $76.7 \pm 0.5$ | |
| + adaptive weight | $69.1 \pm 0.4$ | $74.9 \pm 0.5$ | $77.7 \pm 0.3$ | +0.5 |
| *TextTeacher* (sample) | $71.2 \pm 0.5$ | $75.4 \pm 0.1$ | $78.4 \pm 0.1$ | +1.1 |

tests whether language-based guidance provides complementary regularization that remains effective even when heavy augmentations already diversify the visual input. We evaluate three settings: BorLan's default augmentations, BorLan's augmentation with CutMix added, and the strong 3-augment pipeline (Touvron et al., 2022). As shown in Table 11, BorLan improves moderately with augmentation, and adding our adaptive weighting to BorLan yields a small average gain of +0.5 p.p. In contrast, *TextTeacher* consistently outperforms all BorLan variants, achieving higher accuracy in every regime and delivering a mean improvement of +1.1 p.p. on BorLan *with* adaptive weighting. These results indicate that our method complements strong augmentations rather than competing with them, and that semantic guidance from text continues to provide meaningful signal even when visual transformations are aggressive.

## C.2 Effects of Small Dataset

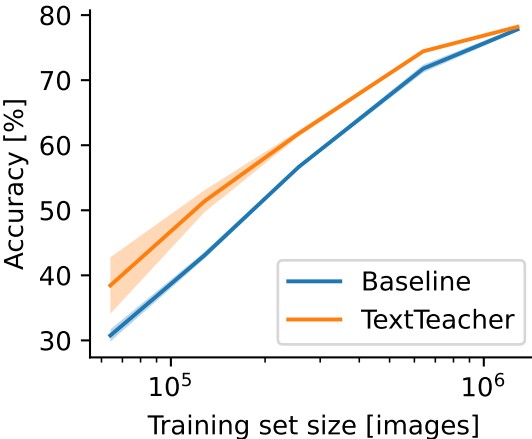

Figure 7: Limited data results for ViT-S with a fixed number of update steps. *TextTeacher* consistently improves accuracy over the vision-only baseline.

To more precisely characterize how our method behaves under varying levels of data scarcity, we measure performance when training on progressively smaller subsets of ImageNet while keeping the total number of optimization steps fixed. This setup isolates the effect of reduced diversity in the training signal rather than reduced compute, allowing us to probe how well each method forms generalizable representations when examples are limited. As shown in Figure 7, accuracy decreases smoothly for both models as the number of training images shrinks, but the relative gap between the baseline and our approach grows consistently in the low-data regime. At moderate subset sizes (e.g., 20–30% of ImageNet), *TextTeacher* already provides several points of improvement, and at the extreme 5% subset it increases accuracy by 7.7 p.p. These results highlight that the auxiliary semantic targets supplied by *TextTeacher* remain informative even when supervision from class labels becomes sparse. In particular, language-derived attributes appear to guide the model toward

more structured feature formation and reduce overfitting, yielding stronger representations exactly where conventional supervised training struggles most.

### C.3 *TextTeacher* at Pretraining and Finetuning

Table 12: Using *TextTeacher* at pretraining time helps (more for larger models) to order the embedding space. During finetuning *TextTeacher* does *not* improve accuracy and might even harm the resulting state if the pretrained model was not aligned with the language guided embedding space.

| Pretrain ImageNet-21k | Finetune ImageNet-1k | ViT-S Accuracy [%] | Delta |
|---|---|---|---|
| No Guidance | No Guidance | 82.0 | |
| No Guidance | $\lambda = 0.5$ const | 77.6 | -4.4 |
| $\lambda = 0.5$ const | No Guidance | 82.3 | +0.3 |
| $\lambda = 0.5$ const | $\lambda = 0.5$ const | 82.3 | +0.3 |

To understand when our language-based guidance is most beneficial, we evaluate its effect when applied during large-scale pretraining on ImageNet-21k and when applied later during finetuning on ImageNet-1k. This experiment probes whether the guidance term serves as a helpful regularizer throughout training or whether its utility depends on the stage at which it is introduced. We run four combinations: guidance enabled or disabled during pretraining, and guidance enabled or disabled during finetuning. For this experiment, we pretrain for 90 epochs on ImageNet-21k and finetune for 30 epochs on ImageNet-1k using the *ViT & others* settings from Table 8, in accordance with Touvron et al. (2022). As shown in Table 12, adding guidance only during finetuning significantly degrades performance (77.6%, −4.4 p.p.), while introducing guidance during pretraining yields a small but consistent improvement (+0.3 p.p.) regardless of whether it is used again during finetuning. This suggests that language guidance is most effective early, when representations are still forming, but can interfere with later specialization. Given that ViT-S is a relatively small model that may already be near capacity under strong supervised pretraining on ImageNet-21k, we expect larger or less saturated models to benefit more substantially from guidance at scale.

## D  Further Analysis

### D.1  Distribution of Caption Length

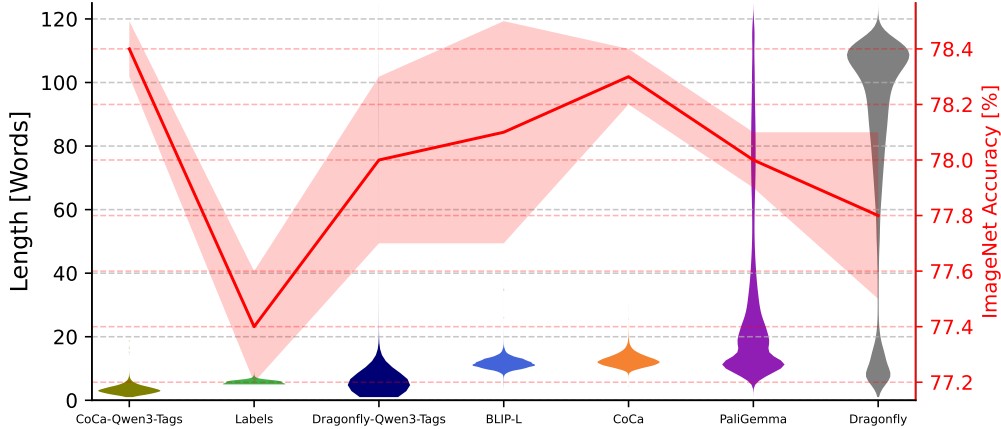

Figure 8: Distribution of the length of image captions for different image captioners compared to ImageNet accuracy when used as text signal in *TextTeacher* with BERT-L as a text encoder.

We analyze whether the amount of textual content provided by different captioners influences the effectiveness of *TextTeacher*. Intuitively, longer captions might supply richer semantic context, but they can also introduce irrelevant details or noise; conversely, very short captions may underspecify the scene. To study this trade-off, we compare the diverse set of captioners from Table 4 and plot their caption-length (in number of words) distributions against the resulting ImageNet accuracy in Figure 8. Captioners are sorted by average length to reveal systematic trends. Note that the reported length for "Labels" includes the prompt template "a photo of <label>". Despite dramatic variation in length, there is no clear trend observable. While the shortest captions (CoCa + Qwen3 tags) are the best and the longest ones (Dragonfly) are the worst, intermediate lengths vary a lot. Overall, the results suggest that caption quality and semantic relevance (which can be increased by using tags instead of sentences) matter more than raw length, and that our method is robust to large differences in caption verbosity. The language signal need not be long, only consistent and semantically aligned with the visual content.

## D.2 CKA Visualization

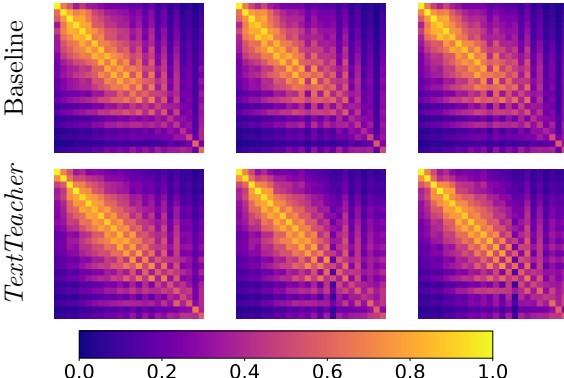

Figure 9: Across run CKA similarity for ViT-S. While different runs are similar in the early layers, *TextTeacher* increases similarity in deeper laters.

To better understand how *TextTeacher* influences model weights, we compute Centered Kernel Alignment (CKA) similarity across independently trained models and visualize the full layer-wise similarity matrices. For each method (Baseline and *TextTeacher*) we train three models with different random seeds and compute CKA for all three pairwise combinations, yielding three matrices per method (Figure 9). These visualizations allow us to inspect the structure and variability of learned representations that are aggregated in Figure 5. Consistent with the trends summarized in Figure 5 of the main text, the Baseline models show substantial variation in deeper layers, whereas *TextTeacher* produces more consistently aligned representations, especially in the later blocks. The higher cross-run similarity visible in the *TextTeacher* matrices indicates that our language-guided auxiliary objectives act as an additional structural prior, encouraging models to converge to more stable representational trajectories.

## D.3 Text Embedding Inversion

To better understand how the classification head and the text head interact in *TextTeacher*, we analyze their alignment by exploring how text-derived embeddings relate to class-discriminative features. Since both heads operate on the same backbone representation via linear maps, the transformation between text-embedding space and class-logit space is analytically tractable. The embedding $\mathbf{z}$ is passed through two linear layers: the class head and the text head.

$$\text{class head:} \qquad \mathbf{z} \mapsto \text{softmax}(A_{\text{CLS}}\, \mathbf{z} + b_{\text{CLS}}) =: \mathbb{P}(\cdot|\mathbf{z})$$
$$\text{text head:} \qquad \mathbf{z} \mapsto A_{\text{TXT}}\, \mathbf{z} + b_{\text{TXT}} =: \mathbf{e}_{\text{TXT}}$$

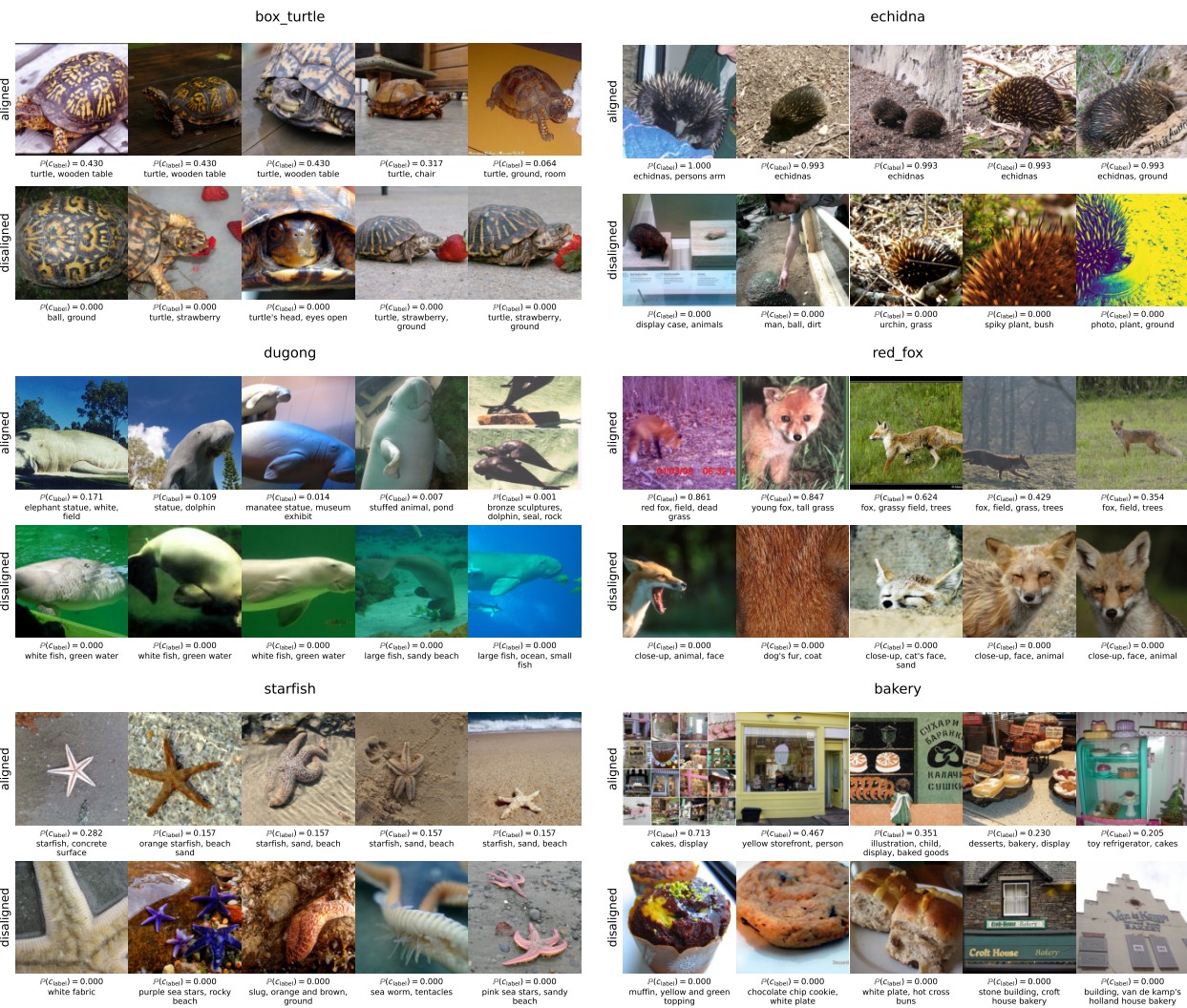

Figure 10: Most aligned and disaligned images for 6 random classes using ViT-B trained with *TextTeacher* for 300 epochs. An aligned image is one where the text captions will output a large signal for a specific class. A disaligned image is an image of a class where the text caption produces a low signal for that class.

We can then compute the left-inverse of $A_{\text{TXT}} \in \mathbb{R}^{d_{\text{txt}} \times d_{\text{emb}}}$ for $d_{\text{emb}} \leq d_{\text{txt}}$:

$$A_{\text{TXT}}^{-1} := \left( A_{\text{TXT}}^{\top} \, A_{\text{TXT}} \right)^{-1} \, A_{\text{TXT}}^{\top}$$
$$\Rightarrow A_{\text{TXT}}^{-1} \, A_{\text{TXT}} = \mathbf{1}$$

Then we can invert the text embeddings to arrive at a class probability distribution:

$$A_{\text{TXT}}^{-1} \left( \mathbf{e}_{\text{TXT}} - b_{\text{TXT}} \right) = \mathbf{z}$$
$$\Rightarrow \text{softmax} \left( A_{\text{CLS}} \, A_{\text{TXT}}^{-1} \left( \mathbf{e}_{\text{TXT}} - b_{\text{TXT}} \right) + b_{\text{CLS}} \right) = \mathbb{P}(\cdot \mid \mathbf{e}_{\text{TXT}})$$

Using this closed-form inverse of the text-head projection, we can map any text embedding back into the shared representation space and evaluate how strongly it activates the classification head. Leveraging this relationship, we conduct a qualitative retrieval experiment: for six randomly selected ImageNet classes, we identify (*i*) images whose text embeddings are maximally aligned with their target class (maximizing $\mathbb{P}(c_{\text{label}} \mid \mathbf{e}_{\text{TXT}})$), and (*ii*) images from that class whose text embeddings are minimally aligned (minimizing $\mathbb{P}(c_{\text{label}} \mid \mathbf{e}_{\text{TXT}})$). The resulting collections in Figure 10 reveal different patterns. In many cases having the

class name in the caption usually results in good alignment (see *echidna*, *red fox*, *starfish*). In other cases, alignment appears driven by broader contextual cues, like *bakery* being aligned with the the concept of baked goods being on display, but *not* with just baked goods or the word "bakery". Similarly, *starfish* is aligned with the word "starfish", but *not* with its synonym "sea star". Additionally, the *box turtle* seems to be aligned with the concept of a turtle being inside. Many disaligned examples stem from captions not representing the image or the class object in an image or being too granular.

