# OpenReview forum: "TextTeacher: What Can Language Teach About Images?"
_TMLR — Accepted by TMLR_

### Review · Reviewer_ixpZ · 2026-02-16

**Summary Of Contributions:**

The submission presents a novel method TextTeacher that provides supervision from text models for ImageNet classification. The images are captioned with a VLM and then the captions are embedded with a text encoder. During training of the vision encoder the classification loss is combined with a contrastive loss that aligns the projected representations with the text embeddings. After training, the text alignment head is discarded and the image encoder can be used exactly the same way as an image encoder that was trained on the classification objective alone.

The authors examine various settings, such as whitening of the text embeddings, different captioners and text encoders, and schedules for the loss weight. The authors present additional insights as to the effect of the added objective to the model weights and to the training dynamics.

**Additional Comments:**

The paper is very well structured, and contains many interesting ablations and experiments that give insights into how the text targets influence the training process.

Questions:

1. What is the difference between "online" and "offline" distillation (Table 2, Appendix B)?
2. Is it correct that "vision distillation" uses a contrastive loss, simply switching the text embeddings with vision embeddings from CLIP/DINOv2? In that case I find the term "vision distillation" somewhat imprecise, and I would prefer a name that more clearly flags that the setup is unconventional.
3. In Table 4 and Figure 8 it seems that CoCa-Qwen3-Tags are shorter than ImageNet Labels. How is this possible? For example in Figure 10 we have single-word labels like "bakery" and they match to 2-7 words tags.
4. Appendix C mentions that "we expect larger or less saturated models to benefit more substantially from guidance at scale". But in Table 1 we see that ViT-L performs worse than ViT-B, DeiT-L performs worse than DeiT-B, XCiT-M performs worse than XCiT-S, even though their gain with TextTeacher is larger. Does it make sense to evaluate TextTeacher in a regimen where the larger models underperform the smaller models?

**Audience:**

Yes

**Audience Explanation:**

I think the submission would be interesting for researchers working on image classification that are also interested in the effect of language supervision on the training process.

**Broader Impact Concerns:**

N/A.

**Claims And Evidence:**

No

**Claims Explanation:**

I am not convinced by the claim in the abstract that "TextTeacher outperforms vision knowledge distillation, yielding more accuracy at a constant compute budget".

**Requested Changes:**

To better support the claim from the abstract (the superiority of the presented method compared to vision distillation), I would like to see two changes:

1. I agree with the authors that compute is an important variable to control for when comparing different methods. On page 10 in "limitations", the authors mention that the labeling with CoCa resulted in 69 hours H100 usage. Additionally to that compute, the method also requires generating tags with Qwen3, and embedding the tags with BERT-L, which should also be accounted for. While I think it is a nice feature to reutilize the compute when running many ablations like in this paper, the initial compute is also required when doing a single training, hence I think it should be counted fully, and not divided by the number of experiments.

2. Table 2 compares "various methods" with TextTeacher. The caption should contain a reference to Appendix B that explains that the other methods are strongly modified to allow a more direct comparison with the text distillation from TextTeacher. It seems that all hyper parameters are tuned for TextTeacher, and then the representations for the contrastive loss are simply swapped from the TextTeacher embeddings to the vision embeddings from CLIP/DINOv2. While I do think that this is an interesting ablation, I don't think this represents sufficient evidence to claim that "TextTeacher outperforms vision knowledge distillation". To support this claim, I would like to see the numbers compared to results from the literature, where the vision distillation methods have been tuned independently to achieve better performance.

---

> ### Author Response · Authors · 2026-03-19
>
> We thank the reviewer for this thorough review of our paper.
>
> ## Compute Requirements
> > the method also requires generating tags with Qwen3
>
> Let us clarify:
> 1. TextTeacher works well with raw captions from a captioning model like CoCa. Tag extraction is optional and serves primarily as an analytical insight into which captions are most effective.
> 2. Tag extraction from captions provides minimal accuracy gain (+ 0.1 p.p., see Table 4).
> 3. Extracting a few keywords from 5-20 word captions is a simple task that could be accomplished with a much smaller model or even a rule-based extraction.
>
> Nonetheless, we agree that all preprocessing compute should be transparently accounted for. We have revised Section 5 and also included a note in Section 4.3.
>
> > the initial compute is also required when doing a single training
>
> Even in the single-run setting, the compute cost is comparable to the overhead of online KD, which adds 50% overhead (~15 min/ep), so the 69 GPU-hours of captioning are amortized within ~275 epochs (less than a standard 300-epoch schedule).
> In practice, model development rarely involves just one run and captions can be shared (we release ours at \<URL\>).
>
> We added a note in Section 4.2 about training compute matched vs. full compute settings.
>
> ## Baselines
> > other methods are strongly modified to allow a more direct comparison with the text distillation from TextTeacher.
>
> We want to clarify that the baseline methods (BorLan, VL2Lite) are not modified. All training hyperparameters are taken directly from DeiT/DeiT-III baselines (Touvron et al. 2021/2022). All method-specific hyperparameters are taken from the original method. For VL2Lite, we apply the projection to vision rather than text embeddings, but these are equivalent (see Appendix B).
>
> > What is the difference between "online" and "offline" distillation?
>
> > Is it correct that "vision distillation" uses a contrastive loss, simply switching the text embeddings with vision embeddings from CLIP/DINOv2?
>
> **Offline guidance** precomputes embeddings for each training image using CLIP/DINOv2. These fixed embeddings are used in place of TextTeacher's text embeddings during training. This setting isolates whether the knowledge source/encoding matters when the distillation mechanism is held constant.
>
> **Online distillation** computes teacher embeddings on-the-fly during training by forward-passing the same augmented views the student receives through the frozen CLIP/DINO teacher. This follows standard KD practice, where the teacher provides view-specific guidance, but incurs ~50% additional compute cost.
>
> We have expanded Appendix B to explain this.
>
> > I would like to see the numbers compared to results from the literature, where the vision distillation methods have been tuned independently
>
> Our baselines align with established methods:
> - CLIP-Embed-KD (Nair, 2024) distills CLIP into classification models using a contrastive loss with constant $\lambda=0.5$, achieving 54.92% on CIFAR-100.
> - CLIP-KD (Yang et al., 2023) finds that MSE loss works for CLIP-to-CLIP distillation. We verify our contrastive approach is competitive in Table 10.
>
> Our goal is to isolate whether text can provide effective guidance to vision models. While more sophisticated vision KD methods exist (e.g. Zhang et al., 2025), they require significant computational resources and architectural modifications. Our online vision baselines achieve performance within expected ranges for the 100-epoch budget: DINOv2 distillation reaches 78.4% vs. 81.1% for officially released DINOv2-S (300 ep).
>
> ## Questions
> > it seems that CoCa-Qwen3-Tags are shorter than ImageNet Labels
>
> The reported length includes prompt template: "a photo of a {label}". We have clarified this in Appendix D.1.
>
> > Appendix C mentions that "we expect larger or less saturated models to benefit more substantially from guidance at scale". But in Table 1 we see that ViT-L performs worse than ViT-B. Does it make sense to evaluate TextTeacher in a regimen where the larger models underperform the smaller models?
>
> We appreciate this observation. The quoted statement refers to Appendix C.4, not Table 1. In C.4, models are pretrained on ImageNet-21k before finetuning on ImageNet-1k. Here, larger models than ViT-S outperform it (see DeiTIII).
>
> For Table 1, the largest models underperform smaller ones. For any training pipeline, there is a size beyond which models no longer outperform smaller ones. TextTeacher can increase this size:
> 1. DeiT-B to DeiT-L: Baseline dramatically drops in performance; with TextTeacher, both are almost on par.
> 2. XCiT-Ti to XCiT-S: Both are on par for the baseline; with TextTeacher XCiT-S outperforms XCiT-Ti.
>
> ## References
> - Nair, "CLIP-Embed-KD: Computationally Efficient Knowledge Distillation Using Embeddings As Teachers", HPEC 2024
> - Yang et al., "CLIP-KD: An Empirical Study of CLIP Model Distillation", CVPR 2023
> - Zhang et al., "Accessing Vision Foundation Models via ImageNet-1K", ICLR 2025

---

### Review · Reviewer_v7i4 · 2026-02-23

**Summary Of Contributions:**

**Summary:**
This paper proposes "Text Teacher," an auxiliary objective designed to improve the performance of vision models by utilizing text information during training, while preserving the unimodal structure of the pure vision model during inference. Specifically, the method uses off-the-shelf Vision-Language Models (VLMs) to generate image captions, summarizes them using a Large Language Model (LLM), and passes them through a frozen text encoder (e.g., BERT) to create text embedding targets. During training, the vision model is guided by adding a contrastive loss that aligns the image features with these text embeddings.

**Key Strengths:**

* The method demonstrates meaningful performance improvements in both accuracy and transfer learning without adding any parameters or computational overhead during inference.


* The paper presents highly practical results showing that the method significantly improves model robustness in environments with severe label noise.



**Key Weaknesses:**

* There is a self-contradiction regarding the claim that the method does not rely on multimodal pretraining. Generating the required captions relies entirely on massive, pre-trained multimodal models (e.g., CoCa, BLIP-L).


* The "compute-matched" comparison with competing methodologies is unfair. It omits the substantial offline preprocessing costs (caption generation, LLM tag extraction, embedding, and covariance calculation for whitening) from Text Teacher's computational budget.


* The theoretical framing around the Platonic Representation Hypothesis (PRH) is overstated. In practice, the method acts as a standard feature-based knowledge distillation regularizer using a well-refined text space.

**Audience:**

Yes

**Audience Explanation:**

Although some claims are overstated, the core methodology is highly practical. The ability to boost a standard vision backbone's performance without degrading inference speed—simply by adding a loss term via offline-built text embeddings—is an attractive proposition for the community. Furthermore, practitioners working in fields where acquiring clean data is difficult (e.g., medical imaging or autonomous driving) will find the strong label noise robustness results highly relevant and interesting.

**Broader Impact Concerns:**

As the authors briefly note in the Limitations section, this methodology relies entirely on text generated by external neural captioning models. Consequently, there is a risk that gender, racial, or cultural biases internalized by these VLM captioners from their massive training corpora could be inadvertently transferred to the vision model's feature space via the text targets. While the authors acknowledge this briefly, I recommend adding a dedicated Broader Impact Statement section to discuss this bias transfer issue—and the limitations of applying this method to specialized, high-stakes domains (e.g., medical or earth observation)—more thoroughly.

**Claims And Evidence:**

No

**Claims Explanation:**

**Explain your answer above***
While the claims regarding performance improvements are well-supported empirically, the claims related to "efficiency" and "avoiding multimodal pretraining" have significant flaws:

1.
**Self-contradictory claims on multimodal pretraining:** The authors repeatedly emphasize that their method avoids the expensive compute of contrastive multimodal pretraining. However, to caption the ImageNet dataset, they utilize massive, data-intensive multimodal models like CoCa and BLIP-L. This is essentially free-riding on the pretraining costs of these VLMs, making the claim that they bypassed multimodal pretraining invalid.


2.
**Unfair compute-matched comparison:** In Table 2, the authors strictly penalize online knowledge distillation methods for the computational cost of running a teacher model alongside the student. However, for Text Teacher, the massive offline preprocessing costs—generating captions (which takes 69 hours on an H100 for CoCa alone), extracting tags with Qwen3, embedding with BERT, and calculating the whitening covariance—are completely excluded from the compute budget. Dismissing this as "negligible overhead" creates an unfair comparison.


3.
**Weak connection to the PRH:** The paper uses the Platonic Representation Hypothesis as a primary motivation. However, the methodology simply uses a refined text embedding space as a soft target for feature alignment. There is insufficient scientific evidence presented to show how this specifically proves or uniquely utilizes the PRH beyond standard regularization techniques.

**Requested Changes:**

**Critical (Must be addressed for acceptance):**

1.
**Tone down claims and acknowledge multimodal reliance:** Remove or heavily revise the claims of achieving results "without resorting to compute expensive contrastive multimodal pretraining" throughout the manuscript. Explicitly acknowledge that the offline caption generation pipeline is a direct product of heavy multimodal pretraining.


2.
**Revise the compute-matched comparison (Table 2):** Correct the 'Compute-matched' comparison in Table 2. Transparently disclose the total offline computation time and resources spent on the entire pipeline—caption generation (CoCa), tag extraction (Qwen3), text embedding, and whitening (BERT)—and factor this into Text Teacher's total training budget to ensure a fair comparison against online distillation methods.


3.
**Refine the PRH framing:** Reduce the excessive emphasis on the Platonic Representation Hypothesis. Reframe the narrative to ground the work more realistically as utilizing the rich prior knowledge of language models as an effective regularizer for vision models.



**Strengthening (Recommended to improve the paper):**

1.
**Address the logical contradiction in scheduling:** The authors conclude that Text Teacher acts as an "early-phase regularizer", but the final experimental setup adopts a constant schedule maintained until the end of training. It would strengthen the paper to add a logical discussion explaining why dynamically decaying the loss to 0 in the later stages (which theoretically aligns better with an early-phase regularizer) did not ultimately yield the best performance.


2.
**Move zero-shot text performance to the main text:** I highly recommend moving the results from Appendix C.1 (Text Encoders as Image Classifiers), which show the text encoder achieves only ~53.1% accuracy on ImageNet, to the main text. This clearly illustrates to the reader that the text signal provided by Text Teacher is not a perfect "answer key" on its own, but rather a structural 'hint' that guides the vision model's representation.

---

> ### Author Response · Authors · 2026-03-19
>
> We thank the reviewer for taking the time to thoroughly review or work. Below, we answer to each of the reviewers points and questions.
>
> ### Multimodal Pretraining
> > The authors repeatedly emphasize that their method avoids the expensive compute of contrastive multimodal pretraining. However, to caption the ImageNet dataset, they utilize massive, data-intensive multimodal models
>
> We want to clarify an important misunderstanding: TextTeacher's *training pipeline* takes captions, images, and labels as input and uses only a unimodal language model to inject textual knowledge into the vision backbone. The pipeline itself requires no multimodal pretraining or optimization of the target model. In our experiments, we use CoCa/BLIP-L to generate captions for ImageNet (which lacks textual annotations). Any source of captions suffices, including human-written descriptions. We chose generated captions to (1) create a large-scale testbed with both captions and labels, and (2) systematically study how caption style and quality affect TextTeacher's performance (see Section 4.3).
>
> That said, we acknowledge that our setup uses and thus benefits from pretrained captioning models. We have revised the manuscript accordingly.
>
> ### Compute Cost
> > The "compute-matched" comparison with competing methodologies is unfair. It omits the substantial offline preprocessing costs (caption generation, LLM tag extraction, embedding, and covariance calculation for whitening) from Text Teacher's computational budget.
>
> We agree that all processing costs should be transparent and have revised the manuscript accordingly. We note that even in the strict single-run setting, TextTeacher's offline costs are comparable to online KD overhead. Online vision KD adds \~50% per-epoch cost (\~15 min/epoch, Table 2), so the 69 GPU-hours of CoCa captioning are amortized within 275 epochs (less than a 300-epoch DeiT schedule) resulting in a net saving of \~6 GPU-hours. In practice, model development involves many runs and the offline costs are paid only once. We also release captions for ImageNet at \<URL\>, making TextTeacher applicable at zero preprocessing cost.
>
> Regarding tag extraction with Qwen3: this step is optional and provides only marginal gains compared to raw captions (+0.1 p.p., Table 4), which directly achieves 78.3%, already outperforming all baselines. We used an LLM, but extracting keywords from short captions is a simple task achievable with much smaller models or rule-based methods.
>
> ### PRH
> > **Weak connection to the PRH**: The paper uses the Platonic Representation Hypothesis as a primary motivation. However, the methodology simply uses a refined text embedding space as a soft target for feature alignment.
>
> We use the PRH as conceptual motivation, not as a claim we prove. The PRH motivated the core question: Can a text-only encoder, without any joint vision-language training, provide useful guidance for vision models?
>
> We agree the method can be viewed as feature-space regularization using text embeddings. However, this view does not oppose the PRH. In fact this regularization only works *because* the text encoder's representations contain semantically relevant information in a geometric structure that the vision model can leverage, which is precisely what the PRH would predict.
>
> We have revised the manuscript to improve the clarity of our claims and contributions, particularly in the introduction.
>
> ### Scheduling
> > Address the logical contradiction in scheduling
>
> The apparent contradiction resolves when examining our jump-schedule results more carefully. The constant schedule outperforms decay-to-zero not because the text signal remains equally important throughout, but because *removing* it (at any point after the critical early phase) introduces an objective shift. As shown in Figure 3, jumps at epochs 70 and 90 degrade accuracy and increase variance compared to the constant on ($\lambda_t \equiv 0.5$) schedule. Mid-training jumps ($t^*=50$) achieve similar accuracy as the constant schedule but notably with higher variance (0.1 p.p. vs 0.03 p.p. for the constant schedule). Thus, the model has adapted its representations to incorporate the TextTeacher signal; removing it introduces instability.
>
> The constant schedule succeeds by simply never imposing this disruption. TextTeacher's primary structural effect is concentrated early (see early to mid-training jumps in Figure 3 (left)), but the continued signal acts as a stable, mild regularizer that, importantly, costs nothing to maintain and avoids the variance introduced by its removal.
>
> We have revised Section 4.5 (now Section 4.6) to include this discussion.
>
> ### Zero-Shot Text Performance
> > Move zero-shot text performance to the main text
>
> We agree this result is valuable in the main text and have moved these results from Appendix C.1 into Section 4.4.
>
> On the suggestion of the reviewer we have added a Broader Impact Statement.

---

### Review · Reviewer_pNn2 · 2026-03-12

**Summary Of Contributions:**

This paper proposes to transfer text embedding structures from language models to vision models, by distillation with an auxiliary head, besides image classification. The text embeddings were obtained from a pretrained CoCa model, then tagged, and then passed through a BERT model and normalized. The experiments show that this auxiliary loss function helps improve performance on imagenet itself, as well as downstream classification tasks. However, the method depends on the pretrained CoCa model to generate captions that contain effectively more text labels, rendering it unclear that the gains come from the language model.

**Audience:**

Yes

**Audience Explanation:**

It is not clear whether TMLR audience might find this paper useful for feature preconditioning. Existing works such as MAE can train ViT models a lot better, with smaller compute requirements, and does not rely on the availability of text embeddings. It is not clear either (discussed above) that TextTeacher is doing better than knowledge distillation, given the fact that it relies on the pretrained CoCa encoder.

However, the extensive ablations in this paper might be interesting to some audience.

**Claims And Evidence:**

No

**Claims Explanation:**

The main concern lies in the main argument of the paper that the distilled knowledge comes from the language model. This argument was almost proved wrong by the experiment in table 4, where the pure ImageNet “labels” plus the language model embedding structure are not sufficient to provide accuracy gains on the main metric evaluated, ImageNet accuracy. Instead, it is the CoCa generated labels or tags, that benefitted from large scale pretraining, that really improved results. The tags are effectively more image-text pairs, e.g. 3.3x, compared with the original ImageNet-1k dataset image-label pairs. It looks plausible to me that the number of tags are the key for improvement, rather than the BERT language model itself. This view seems supported by section C.3 where we see that the method improves data efficiency by a fixed ratio.

In addition to the main concern described above, the baselines in this paper are particularly weak. For example, in Table 1, the gains are only pronounced when the baselines are weak, or more specificall, overfitting e.g. with ViT models. On improved recipes, such as DeiT-B, the gains are quite marginal (+0.4%). The DeiT-L results are less convincing since they are much worse than the DeiT-B and are not produced in the original paper.

Finally, given the fact that the CoCa model was used to caption the images, it is good to understand how this CoCa model is doing when it is evaluated on ImageNet, e.g. zero-shot, linear eval, fine-tuned, and how it works when it is distilled to a smaller ViT model.

**Requested Changes:**

One could show that, on top of the CoCa caption tags, an existing CLIP training (i.e. without the pretrained BERT initialization) is not sufficient to improve results, over ImageNet. One could also show that if a model is pre- or co-trained on a subset of ImageNet-21k (e.g. using 3M-4M image label pairs, similar to the number of tags in this paper), how the results compare with TextTeacher.

---

> ### Author Response · Authors · 2026-03-19
>
> We thank the reviewer for the detailed feedback.
>
> ## Where do the gains come from?
> > It is the CoCa generated labels or tags, that really improved results.
>
> This interpretation is not supported by our ablation evidence, instead *both* the textual content *and* its encoding matter:
> - In Table 5 (Text Encoder Ablation): Keeping captions identical, swapping BERT for weaker text encoders significantly degrades gains, demonstrating that the *representation* of the knowledge is important.
> - In Table 2: This textual knowledge source, is more informative for the student than directly distilling from a pretrained image encoder (even CoCa).
>
> In Table 4 ImageNet class labels alone do not improve accuracy, since single class labels produce identical text embeddings for all images of a class.
> While the additional knowledge does come from the text, it must be encoded into a form that can effectively enhance the vision model.
>
> >  It looks plausible to me that the number of tags are the key for improvement, rather than the BERT language model
>
> Note, that tags (fed to the model as a single string) are extracted from captions and thus only contain a subset of information the caption itself contained. If the sheer number of text-image pair equivalents were the mechanism, one would expect the captions to outperform the tags, however we observe that the tag-strings slightly outperform the captions. Additionally, the extra information alone is far from sufficient to solve the classification task (see new Section 4.4).
>
> ## Baselines and Improvements
>
> > the baselines in this paper are particularly weak.
>
> We respectfully disagree. DeiT, Swin, and NextViT are strong, well-established baselines that reach >80% on ImageNet. DeiT is the de-facto standard baseline for supervised training of vision transformers (Touvron et al. 2022; Bao et al. 2022; Rangwani et al. 2024; Montrezol et al. 2026).
>
> > On improved recipes, such as DeiT-B, the gains are quite marginal
>
> DeiT-B at 81.8% is already a well-optimized recipe and the +0.4 p.p. gain is not negligible for a large-scale dataset. Notably, it's the same effect as knowledge distillation from DINOv2-L, but 33% faster. Moreover, we observe $\approx 1.0$ p.p. gains on downstream tasks.
> Note also the other architectures where gains are larger, like XCiT-M, demonstrating that TextTeacher is consistently effective.
>
> > given the fact that the CoCa model was used to caption the images, it is good to understand how this CoCa model is doing when it is evaluated on ImageNet
>
> We thank the reviewer for this suggestion. We investigate knowledge distillation from CoCa directly (added to Table 2), without going to text embeddings first. Results are close to distillation/guidance from CLIP and DINO and worse than TextTeacher at the same compute cost. Thus, the gain is not only coming from CoCa, but also from the language-informed embedding space.
>
> ## Other Training Settings
> > Existing works such as MAE can train ViT models a lot better, and does not rely on the availability of text embeddings.
>
> TextTeacher and MAE are complementary rather than competing approaches. MAE is a self-supervised pretraining method, while TextTeacher is an augmentation, not requiring the overhead of pretraining of the target model.
>
> > It is not clear that TextTeacher is doing better than knowledge distillation, given the fact that it relies on the pretrained CoCa encoder.
>
> Standard knowledge distillation from CoCa transfers CoCa's *visual* representations to the student. TextTeacher instead transfers *linguistic* representations. As discussed above, TextTeacher outperforms visual knowledge distillation from CoCa.
>
>  > One could show that, on top of the CoCa caption tags, an existing CLIP training is not sufficient to improve results
>
> Our existing ablations (Table 5) already address the core question of whether a pretrained text encoder matters: The quality of the text representation space, not merely the presence of a contrastive objective, matters. CLIP-training would require jointly optimizing a text encoder during training; precisely the computational overhead TextTeacher avoids.
>
> > One could show that if a model is pre- or co-trained on a subset of ImageNet-21k, how the results compare with TextTeacher.
>
> While this would likely yield improvements, we note: (1) it provides *additional images* with additional visual information, while TextTeacher uses only the original ImageNet-1K images; (2) it requires substantially more compute than our setup.
>
> ## References
> - Bao et al. ``BEiT: BERT Pre-Training of Image Transformers'', ICLR 2022
> - Rangwani et al. ``DeiT-LT Distillation Strikes Back for Vision Transformer Training on Long-Tailed Datasets'', CVPR 2024
> - Montrezol et al, ``Decoding vision transformer variations for image classification: A guide to performance and usability'', MLWA 2026
> - Touvron et al. ``DeiT III: Revenge of the ViT'', ECCV 2022

---

### Author Response · Authors · 2026-03-19

We thank all reviewers for their thoughtful and constructive feedback. We are encouraged that reviewers found the paper well-structured (ixpZ), the ablations insightful (ixpZ, pnN2), and the core idea of leveraging text as a training signal for vision models interesting and relevant (v7i4).

We address each reviewer's concerns in detail in our individual responses. Here we summarize the key shared themes:

## Additional compute cost
Reviewers raised concerns about preprocessing overhead (captioning, embedding). We emphasize that this is a *one-time, offline* cost: captions and embeddings are computed once and cached. For ImageNet, the full pipeline takes ~69 GPU-hours on a single H100, which is comparable to a single training run. This cost amortizes across all subsequent experiments, and during training itself, TextTeacher adds *zero* computational overhead since it operates on precomputed embeddings. We will release all captions and embeddings, making TextTeacher applicable to ImageNet at no preprocessing cost.

## Reliance on pretrained multimodal models
Several reviewers questioned whether TextTeacher's gains stem from multimodal knowledge embedded in the captioner (CoCa). We want to clearly disentangle two claims:
1. Our *pipeline* uses a VLM for convenience, but any caption source works, as the method is agnostic to caption origin.
2. The *signal* TextTeacher provides is purely linguistic, not multimodal. To validate this, we show that directly distilling from CoCa's visual encoder underperforms TextTeacher (results added to Table 2), confirming that the benefit comes from the structure of language, not from vision-side knowledge leaking through the captioner. The text encoder (BERT) is language-only and was never trained on images.

We have revised the manuscript to address all reviewer suggestions and are happy to discuss any remaining concerns.

---

### Decision · Action_Editor_Rpz1 · 2026-04-27

**Recommendation:** Accept with minor revision

**Additional Comments:**

Most of the claims are well-supported in the revised manuscript. However, Reviewer v7i4 points out a remaining concern: the claim that "any caption source works" remains under-validated at the ImageNet scale. Specifically, labels alone do not outperform the baseline (Table 4), and no experiments utilizing human-written or rule-based captions are provided.

While the current acknowledgment regarding VLM dependence is sufficient for acceptance, the authors are required to further soften the "any caption source works" claim in the camera-ready version to accurately reflect the scope of the provided evidence.

**Audience:**

Yes

**Audience Explanation:**

The revised submission meets TMLR's acceptance criteria: the claims are supported by the evidence, and the findings are of practical interest to practitioners training unimodal vision models under budget or latency constraints. As a result, there should be substantial interest in this work within the TMLR readership.

**Claims And Evidence:**

Yes

**Claims Explanation:**

This paper presents TextTeacher, a framework that utilizes text embeddings during image classification training while deploying only the vision model during inference. Specifically, an off-the-shelf captioning model (CoCa) is employed to generate image captions, which are then summarized by a large language model (Qwen3). The resulting tags are fed into a frozen text encoder (BERT) to create text embeddings. During training, the vision model is trained using a standard cross-entropy loss alongside a contrastive loss that aligns the image features with the extracted text embeddings.

Reviewers v7i4 and ixpZ acknowledge that the main claim—that the semantic knowledge learned by a language model can efficiently train a vision model—is empirically well-supported. The method demonstrates consistent improvements over baselines, though the reviewers noted these improvements diminish in the DeiT-S and DeiT-B configurations. While Reviewer pNn2 expressed concern regarding whether the distilled knowledge originates purely from the language model, the Action Editor agrees with Reviewer v7i4 and ixpZ that the added comparison against direct distillation from CoCa's visual encoder (Table 2) and the text-encoder ablation study (Table 5) sufficiently demonstrate that the performance gains are not merely attributed to CoCa's visual representations leaking through the captions.

Additionally, Reviewer v7i4 confirmed that the authors have properly rephrased their claims regarding a "fair compute comparison" and "the connection to the Platonic Representation Hypothesis" in the revised version.